



# Elevated melt causes varied response of Crosson and Dotson Ice Shelves in West Antarctica

David A. Lilien[1,2], Ian Joughin[1], Benjamin Smith[1], and David E. Shean[1,3]

[1]Polar Science Center, Applied Physics Lab, University of Washington, Seattle, Washington, USA.
[2]Department of Earth and Space Sciences, University of Washington, Seattle, Washington, USA.
[3]Department of Civil and Environmental Engineering, University of Washington, Seattle, Washington, USA.

*Correspondence to*: David Lilien (dal22@uw.edu)

**Abstract.** Crosson and Dotson Ice Shelves are two of the most rapidly changing outlets in West Antarctica, displaying both significant thinning and grounding-line retreat in recent decades. We used remotely sensed datasets to investigate the processes controlling their changes in speed and grounding-line position over the past 20 years. We combined these observations with inverse modeling of the viscosity of the ice shelves to understand how weakening of the shelves affected this speedup. These ice shelves have lost mass continuously since the 1990s, and we find that this loss is primarily a result of melt beneath Dotson. High melt rates persisted over the period covered by our observations (1996-2014), with the highest rates beneath areas that ungrounded during this time. Grounding line flux exceeded basinwide accumulation by about a factor of two throughout the study period, consistent with earlier studies, resulting in significant loss of grounded as well as floating ice. The near doubling of Crosson's speed in some areas during this time likely is the result of weakening of its margins and retreat of its grounding line. This speedup contrasts with Dotson, which has continued to move slowly despite high, increasing melt rates near its grounding line. Our results indicate that changes to melt rates began before 1996, and suggest that observed increases in melt in the 2000s compounded an ongoing retreat of this system. Advection of a channel along Dotson, as well as the grounding-line position of Kohler Glacier, suggest that Dotson experienced a change in flow around the 1970s, which may be the initial cause of its continuing retreat.

## 1 Introduction

Glaciers in the Amundsen Sea Embayment are susceptible to instability and are the dominant source of sea level rise from Antarctica (Shepherd et al., 2012). Observations (Rignot et al., 2014) and modeling (Favier et al., 2014; Joughin et al., 2010, 2014) suggest that collapse is potentially underway on Pine Island and Thwaites Glaciers, the largest in the region. The neighboring Smith, Pope, and Kohler Glaciers thinned at 9 ma$^{-1}$ just upstream of their grounding lines from 2003-2008, outpacing the mass loss of those larger catchments in relative terms (Pritchard et al., 2009). The lower reaches of these glaciers, which discharge to Crosson and Dotson ice shelves (for brevity, we hereafter refer to these ice shelves simply as Crosson and Dotson), doubled their speeds, as their respective grounding lines retreated by as much as 35 km in places from 1996 to 2011 (Rignot et al., 2014). During this same period, portions of Crosson sped up at a rate comparable to the grounded areas upstream, while Dotson remained at near-constant velocity



(Mouginot et al., 2014). Thus, this pair of ice shelves with differing speedup but similar incoming fluxes and neighboring catchments provides an ideal area in which to study the processes controlling ice-shelf stability.

Ice shelves affect upstream dynamics by exerting stresses at the grounding line. For ice masses, the driving stress, $\tau_d$, is resisted by basal drag, $\tau_b$, longitudinal stress gradients, $\tau_L$, and lateral stress gradients, $\tau_W$. To maintain force balance, any reduction in resistance must be compensated by increases in other forces. This adjustment generally causes the glacier to speed up to increase strain-rate dependent stresses ($\tau_L$ and $\tau_W$) within the ice. Reduction in ice-shelf resistance (also called buttressing) can be caused by thinning (e.g. from increased basal melt) or changes in extent (e.g. from shelf breakup or increased calving). Weakening of ice-shelf margins, due to heating, fabric formation, or crevassing or rifting (through-propagating crevasses) also effectively reduces the lateral stress gradients and reduces buttressing (Borstad et al., 2016; Macgregor et al., 2012). Despite either mechanical or rheological "weakening", marginal ice can still exert a greater resistance to flow as a glacier speeds up due to the dependence of stress on strain rate.

Areas of compression and tension in ice shelves have been identified as key to shelf stability (Doake et al., 1998; Sanderson, 1979). There is generally a region of tensile stress near the calving front, while upstream of this region, some components of the stress tensor are in a compressive regime; the boundary between the two is often arcuate, and thus is referred to as a "compressive arch." Changes seaward of the arch have little effect on upstream dynamics since the shelf is essentially freely spreading in this region. Changes at or upstream of the arch, such as rifting, or other weakening, can cause widespread speedup because compression, and thus resistance to upstream flow, is reduced.

The dynamics and stability of ice shelf/ice stream systems are also strongly controlled by grounding-line position. Retreat of the grounding line along a landward-sloping (retrograde) bed causes imbalanced flux due to the nonlinear dependence of ice flux on ice thickness in what is termed the marine ice-sheet instability (Weertman, 1974). Theoretical work has shown that, once perturbed, a grounding line on a retrograde bed will retreat until it reaches a prograde slope (Schoof, 2007). Additionally, ungrounding increases the sensitivity to melt by exposing more ice to the ocean (Jenkins et al., 2016). In the case of a retrograde bed, the deepening of the grounding line caused by ungrounding also increases melt both because the melting point decreases with depth and because deeper ocean waters are generally warmer in these systems. Retreat of the grounding line thus causes both imbalanced flux at the grounding line and allows elevated melt rates beneath floating ice. Theoretical arguments provide some insight into the importance of rifting, ungrounding, and loss of buttressing in idealized conditions, but their role in controlling flow during complex evolution of real ice streams is not fully known. Here we investigate the role of these processes in recent speed and thickness change of Crosson, Dotson and their tributary glaciers.

## 1.1 Study Area

Pope Glacier and the eastern branch of Smith Glacier together feed Crosson, while the western branch of Smith Glacier and Kohler Glacier feed Dotson (Figure 1). The terminology for these tributaries has varied in the literature, and we





adopt the names used in Scheuchl et al. (2016) for consistency and clarity. The largest grounding line retreat (>2 km a$^{-1}$) occurred in the area where the branches of Smith Glacier flow together before splitting into their respective shelves (Figure 2). The grounding-line position of Smith Glacier was relatively stable from 1992 to 1996, but it then retreated substantially in the 18 years following (Rignot et al., 2014; Scheuchl et al., 2016). By contrast, the grounding line of

Kohler experienced limited retreat (~0.2 km a$^{-1}$) from 1996 to 2011 and subsequently re-advanced to near its 1996 position by 2014 (Scheuchl et al., 2016). Though data are unavailable for intermediate dates, Pope Glacier's grounding line retreated by 11.5 km (0.64 km a$^{-1}$) from 1996 to 2014 (Scheuchl et al., 2016). The ice generally thinned most strongly over the areas that ungrounded and up to ~15km upstream (contours in Figure 2). This correspondence between thinning and retreat is spatially coincident with speedup, which was strongest over grounded portions of

Smith Glacier. Ice speeds in the strongly thinned, grounded area and downstream on Crosson peaked in 2009 and 2010 then declined slightly by 2014. There were also significant changes in the extent of Crosson and thus in the amount of contact with its sidewalls and with the tongue of Haynes Glacier (Figure 1b) through this period. Digitized shelf-front positions throughout the ASE indicate extensive rifting of Crosson, resulting in detachment from both of its margins, as well as breakup of the Haynes Glacier tongue from 1984 to 2004 (Macgregor et al., 2012), which could

have reduced the ability of the shelf to transmit resistive stresses. No similar changes were observed over Dotson.

The extensive and synchronous changes in the ASE have mainly been attributed to changes in basal melt (e.g. Joughin et al., 2012 and references therein) caused by warming ocean water or increased intrusion of warm, salty circumpolar deep water (Jenkins et al., 2010; Pritchard et al., 2012; Thoma et al., 2008). Recent work has used radio echo sounding (RES) data to infer thinning rates of 70 m a$^{-1}$ (Khazendar et al., 2016), but the extent of these estimates is limited in

space and time, and converting to a melt rate requires assumptions about the magnitude of dynamic thinning in the area, and about the steady-state melt rates. Gourmelen et al. (2017) used thinning rates from Cryosat-2 to obtain a more spatially complete record of melt rates over Dotson, particularly focusing on melt rates in a channel beneath the shelf (Figure 1), and found that melt rates in the channel are much higher than the surrounding ice. Observations of ocean temperature in front of Crosson are sparse due to persistent sea ice and ice mélange, though there are

observations elsewhere in nearby Pine Island Bay (Jacobs et al., 2012; Wåhlin et al., 2013). Recently, the front of Dotson has been better instrumented, and observations show inflow at the eastern margin of the shelf and outflow at the western margin (Ha et al., 2014; Miles et al., 2015), indicating clockwise circulation beneath the shelf. The inflow is persistently warm at depth while the outflow is cooler and fresher, indicating the presence of meltwater. Available

data suggest that in 2006, water in Pine Island Bay, which may access Crosson, was about 0.7°C warmer, and less variable in temperature, than the inflow to Dotson (Jacobs et al., 2012; Walker et al., 2007). However, attempting to infer melt rates beneath the shelves from these few measurements is not straightforward. Furthermore, sub-ice-shelf bathymetry has only recently become available (Millan et al., 2017), limiting model-based analyses of ocean circulation beneath these shelves.


Previous ice-flow modeling, which excluded the floating ice, suggested that these glaciers will continue to retreat even in the absence of a change in forcing (Goldberg et al., 2015). Since changes in the ASE are thought to be ocean-



forced, however, understanding of processes over the floating ice and at the grounding line are key to explaining recent behavior and to any prediction of future behavior. Here we use remote-sensing observations of ice velocities, surface elevations, and ice-bottom elevations of these ice shelves through time to determine the partitioning of their mass loss and to constrain snapshot inversions of their viscosity. We use the history of melt, terminus position, and shelf

viscosity to understand the causes of changes to these shelves' mass balance and speed.

## 2 Data

Before describing the calculations of mass loss and modeling, we first summarize the data used in this study.

### 2.1 Velocity

Surface velocities were obtained from both synthetic aperture radar (SAR) and optical satellites. The SAR velocities

come from the European Remote-sensing Satellites (ERS-1 and -2) for 1996 and the Advanced Land Observation Satellite for 2006-2010. These SAR data were processed using a combination of interferometry and speckle tracking (Joughin, 2002). We used feature tracking of Landsat-8 imagery to obtain velocities for the 2014-2015 austral summer. Velocity data for 2006 and 2011 are part of the NASA Making Earth System Data Records for Use in Research Environments (MEaSUREs) data set (Mouginot et al., 2014). Errors range from a few meters per year over

slower grounded ice to >100 m a$^{-1}$ over Dotson in 1996 due to the short satellite repeat period and errors introduced by tidal displacement.

### 2.2 Surface elevations

To estimate past surface elevations, we began with elevations from a high-quality digital elevation model (DEM) with elevations relative to the EGM2008 geoid. This reference surface is a mosaic DEMs created through processing of

stereo imagery from the DigitalGlobe WorldView/GeoEye satellites (Shean et al., 2016), using imagery spanning 2010-2015. Most of the stereo pairs used to create this mosaic come from the 2010-11, 2012-13, and 2013-14 austral summers, so we assign the mosaic an approximate timestamp of January 1, 2013. This surface elevation product is posted at 32m, and we estimate the error to be ±1.0 m. To find surface elevations in different years, we added the observed thinning rates, as discussed below, to this reference surface. Because there are published estimates of

thinning rates over the ice shelves (Paolo et al., 2015; Shepherd et al., 2004), we used these rates for floating ice and addressed grounded ice separately.

Over grounded ice, we used thinning rates derived from a combination of points from ICESat-1 and the Airborne Topographic Mapper (ATM) as well as DEMs produced from stereo pairs of Worldview imagery; details of how this

time series was produced can be found in Appendix A of Goldberg et al. (2015). These observations of surface elevation span 2003-2015, necessitating extrapolation to determine elevations at the beginning of our study period. To extend to 1996, we fit a quadratic function to each pixel of the 2003-2014 elevation change record; the use of a quadratic description of the thinning follows previous work (Wingham et al., 2009). We then used these thinning functions to calculate surface elevations over these glaciers during 1996-2002. The results from this method more





closely match the available ICESat-1 data than using the elevation and thinning rate from 2004 alone, as was done in Mouginot et al. (2014).

Over floating ice, we used 1994-2012 thinning rates derived from satellite radar altimetry data by Paolo et al. (2015).

The results from Paolo et al. (2015) show little spatial variability over these particular shelves, and so we used the thinning rates for the middle of each shelf where low surface slopes should lead to the smallest errors. These values were 3.1 m $a^{-1}$ for Crosson and 2.6 m $a^{-1}$ for Dotson. Because of relatively high surface slopes and the 30-km resolution of these estimates, the thinning rates are not accurate fully to the margins of the shelves, and there are ~5 m $a^{-1}$ differences in our thinning function upstream and downstream of the grounding line. The thinning rates on the

grounded ice are more accurate since they lack tidal effects, so we smoothed the thinning over the shelves for 10 km downstream of the grounding line to preserve continuity and reasonable surface slopes.

### 2.3 Ice-bottom Elevations

We use a 1-km bed elevation dataset generated from all available airborne RES data with an anisotropic interpolation routine that weights measurements along flow more heavily than those across flow; details can be found in the

supplementary materials to Joughin et al. (2014) and Medley et al. (2014). This method reduces many of the artifacts that can occur when interpolating sparse ice thickness measurements, while avoiding making any assumptions about the present state of balance (e.g. assumptions for mass conservation methods (Morlighem et al., 2011)). We estimate bed elevation errors of ~50 m for the study area.

Over floating ice, we used the surface elevation for a given year and an assumption of hydrostatic equilibrium to calculate the elevation of the lower ice surface. In doing so, we first determined the firn-air content by choosing the value that minimizes the misfit in ice thickness between coincident surface elevation (converted to ice thickness by subtracting the air content and assuming floatation) and ice thickness measurements; elevation and thickness measurements were taken from the ATM and Multichannel Coherent Radar Depth Sounder (MCoRDS) flown on

NASA's Operation IceBridge. The ice thickness measurements from the MCoRDS radar are calculated from the two-way travel time of the radar using a wave speed for pure ice, but the air content in the firn changes the radar wavespeed. In our minimization, we convert the thickness to a travel time, reversing the MCoRDS processing methods, then convert the travel time back to thickness using a wavespeed that accounts for the air content of the firn. This approach provides point measurements of firn-air content along tracks separated by several kilometers, which we then gridded

and subtracted from the surface elevation, resampled to 400-m posting, before applying the assumption of hydrostatic equilibrium to compute the ice bottom elevation. Residuals between the ice thickness obtained from this procedure and direct measurements of ice thickness from MCoRDS have an RMS of 22 m. This error is larger than the ~10-m crossover precision in the MCoRDS data in this area, but comparable to the absolute accuracy of those measurements, which include error from uncertainty from the dielectric constant, limited sampling rate, and uncertainty in picking

the reflector.



### 2.4 Ice-front positions

We digitized the shelf-front positions of Crosson and Dotson from 2012 to 2016, extending the earlier 1972-2012 record (Macgregor et al., 2012). The shelf-front positions were traced on Landsat-8 imagery, using the panchromatic data (band 8). Shadowing and brightening of steep areas causes an approximately four-pixel (±60m) nominal
uncertainty in the front position, though actual error is ~500m on Crosson due to ambiguity in discriminating between ice front and persistent mélange.

### 3    Methods

We derived annual flux, melt, and calving rates through time using the observed velocity and ice thickness. We compared these values to flux at the grounding line and accumulation on the shelves to determine changes in the ice-
shelf mass balance. Since stresses in the ice cannot be measured remotely, we performed snapshot inversions for viscosity using a numerical model to quantify how changes in velocity and geometry affected the stress balance and strength of Crosson and Dotson.

### 3.1 Flux and melt calculation

Following earlier work (Mouginot et al., 2014), we used the average of five closely spaced gates to calculate inflow
and outflow fluxes along the shelf boundaries. For grounding-line flux, gates were drawn at 2-km intervals beginning just upstream of the 2014 grounding line. Onto each gate profile, we interpolated surface and bed elevations and the component of velocity orthogonal to the gate. We filled gaps < 3km wide via 2-D linear interpolation of surrounding velocities, and filled larger gaps by 1-D linear interpolation between the velocities at that point the two closest years with sufficient coverage. Where we have velocity data, we take the error to be the formal value, and where we have
interpolated we estimate it to be the difference between the interpolated value and the values at neighboring years.

We calculated melt rates over different portions of the shelves to understand the spatial distribution of melt. Basal melt is effectively a downward flux of ice out of the shelf bottom, so to calculate melt we divided the shelves into polygons (Figure 3a), calculated the incoming and outgoing flux for each polygon, then used mass conservation to
determine the melt rate. We used the average flux through five parallel gates on each side of the polygons to reduce errors in the horizontal fluxes. Where possible, we located the upstream sides of the polygons over ice that was grounded in 1996 to reduce errors in the velocity caused by tidal variation over the short repeat cycle employed by ERS during this period. We estimate the error in the margins of floating ice, which propagates to error in the area, to be 5%; this error does not affect our calculation of melt for individual polygons, but does affect our estimate of total
melt on each shelf. To relate the flux into and out of a polygon to the melt rate, we integrated the mass change of the polygon and applied divergence theorem:

$$\int_{\Omega} \dot{m}_b \, dA = \int_{\Omega} \left( -\nabla \cdot \boldsymbol{Q} + \dot{m}_s + \frac{\partial H}{\partial t} \right) dA = -\oint_{d\Omega} \boldsymbol{Q} \cdot \hat{\boldsymbol{n}} ds + \int_{\Omega} \left( \dot{m}_s + \frac{\partial H}{\partial t} \right) dA \qquad (1)$$





where $\dot{m}_s$ is the basal melt rate, $\boldsymbol{Q}$ the depth-integrated flux, $\dot{m}_s$ the surface mass balance (SMB, water equivalent), $\frac{\partial H}{\partial t}$ the thickness change, $\Omega$ a polygon with boundary $d\Omega$, $\hat{\mathbf{n}}$ the unit normal to the boundary, $\mathbf{A}$ the area of the polygon, and $\mathbf{s}$ the distance along the boundary. To determine melt rates from the flux balance of each polygon, we used the rate of thickness change from Paolo et al. (2015) and SMB from RACMO2.3 (Van Wessem et al., 2014). For the

SMB, we use the annual mean for 1979-2013, and, following Depoorter et al. (2013), we take the error in the SMB to be ±28%. The assumption of a spatially uniform thinning rate for each shelf introduces additional error that is difficult to quantify, but without more measurements of surface change this difficulty is unavoidable. We computed mass loss rates for both ice shelves and catchments. The former is useful because the ablation on the shelves is greater than the grounding line flux, causing them to lose significant mass annually, while the latter gives an indication of the system's

contribution to global sea level rise and is useful for comparison to other studies.

### 3.2 Modeling

We used a diagnostic model implemented in Elmer/Ice (Gagliardini et al., 2013; Zwinger et al., 2007) to infer ice-shelf viscosity. Elmer/Ice is an open-source, finite element software package capable of solving the full Stokes equations in three dimensions, or lower-order approximations to ice flow such as the shallow-shelf equations

(Gagliardini et al., 2013; MacAyeal, 1989). We used a three-dimensional, full-Stokes model with separate domains for grounded and floating ice. While the assumptions of the shallow-shelf equations are likely applicable at least to the floating portion of our domain, we implemented the Full-Stokes model to accurately capture the effects of ~10 pinning points beneath these shelves and avoid approximations to the stress state.

We performed multiple diagnostic runs in which we varied the model geometry to reflect different years, which allowed us to infer properties through time as a series of snapshot inversions. The upstream margins of the model domain were located at the divides in ice flow, determined from InSAR velocity measurements. We found the downstream margin using the digitized shelf-front positions described above. The grounding-line positions from (Rignot et al., 2014; Scheuchl et al., 2016) were used to determine the boundary between the grounded and floating

domains for each year. We used these horizontal extents, in conjunction with the different surface and bed DEMs described above, to create a suite of three-dimensional model meshes using the software GMSH (Geuzaine and Remacle, 2009). These horizontally unstructured meshes have a resolution of ~300-m over floating ice. The meshes have eleven vertical layers, with six of these in the bottom third to capture the zone of maximum internal deformation; there is little internal deformation over most of the floating domain, but this concentration of layers is useful for

capturing temperature gradients in the ice column and dynamics around pinning points.

To obtain an initial estimate of viscosity for the enhancement factor inversions, we first determined a temperature profile in the ice. This initial estimate was made using a steady-state thermo-mechanical model in Elmer/Ice. The model solves the Stokes equations to determine advection, with ice viscosity as a function of temperature, and solves

an advection-diffusion equation for heat using limiters to prevent ice from going above the pressure-melting point (Zwinger et al., 2007). Strain heating and frictional heating at the bed were both included in this temperature model.





For boundary conditions, we used surface temperatures from RACMO2.3 (Van Wessem et al., 2014) and geothermal heat flux estimates from geomagnetics (Maule et al., 2005). We first ran the temperature model over grounded ice; the downstream temperatures from the grounded ice were then used as the upstream boundary condition for a similar model of the floating ice.

After obtaining an initial viscosity from the temperature model, we used inverse methods to infer the enhancement factor that produced modeled velocities that best match observations. The enhancement factor alters the viscosity of the ice relative to that predicted solely by modeled temperature. It would be possible to use the model to find a depth-variable enhancement factor, but, to have the number of degrees of freedom in the inversion match the number of

observables, we instead determined a single depth-independent value. We began with a profile of viscosity with depth from the temperature model, and used adjoint methods (MacAyeal, 1993; Morlighem et al., 2010) to infer an enhancement factor that results in the best fit to observations for each column of ice. Because we were inferring variations along shear margins with sharp transitions in velocity, temperature, and rheology, we did not apply regularization to the inversion for the enhancement factor, and simply minimized the areally integrated misfit between

observed and modeled velocity. The lack of regularization may have concentrated the weakening or strengthening into smaller areas than would have been found with regularization, but this choice should not have affected the overall spatial pattern.

For boundary conditions, we used observed surface velocity for all depths over lateral margins (including shear

margins, calving front, and grounding line). For the basal boundary, we used zero basal shear stress over floating ice and basal shear stress equal to half the driving stress over the pinning points observed from SAR grounding lines.

### 3.3 Stress Changes

The location of changes in rheology relative to the compressive arch is an important factor in determining whether those rheological changes affect the broader flow patterns of the shelf. To determine the location of the compressive

arch on Crosson and Dotson, we used the modeled stress field to calculate the stress associated with shelf spreading, $\tau_L = 2\bar{\tau}_{xx} + \bar{\tau}_{yy}$ where $\bar{\tau}_{xx}$ and $\bar{\tau}_{yy}$ are the depth averaged deviatoric stresses along and across flow. We chose this criterion because where it is compressive it shows resistance to spreading upstream and where it is tensile it indicates a spreading shelf not resisting flow. This approach differs from previous literature, which used the second principal stress to identify compressive arches; other studies choose that criterion in part because the first principal stress is

consistently tensile (Doake et al., 1998), which is not the case on Crosson and Dotson, and so ignoring the first principal stress misses an important aspect of the stress balance here.



## 4   Results

### 4.1 Flux and melt

Figure 3 shows the components of the flux balance of Crosson and Dotson throughout the study period.  The three columns for each year show the outgoing flux, incoming flux, and loss, respectively.  The outgoing flux is partitioned

into the melt on different portions of the shelves and the calving flux. The outgoing flux from some polygons is equal to the incoming flux of others; when adding all the melt together the flux across these internal boundaries cancels, leaving the total melt on the shelf.  The incoming flux consists of accumulation on the shelf and the flux across the grounding line; we partition the grounding-line flux into a steady-state amount (i.e. the accumulation in the catchment upstream) and any additional amount entering the shelves in each year.  The difference between the incoming and

outgoing fluxes for the shelves provides an estimate of net mass loss from the shelves, while the difference between upstream accumulation and grounding-line flux yields net loss from grounded ice.

The grounding-line fluxes into Crosson and Dotson both exceeded their upstream accumulation in 1996.  By 2014, the fluxes across the grounding lines increased by 30% and 60%, respectively.  Increases in outgoing flux (calving

and melt) outpaced increases in grounding-line flux for both Crosson and Dotson, leading to loss of shelf volume. Due to Crosson's speedup, its calving-front fluxes increased from 1996 to 2010 then declined slightly to 2014, while the calving-front flux from Dotson declined due to both thinning and slight slowdown.  Total melt beneath each shelf also increased from 1996 to 2010, and declined slightly to 2014.  The melt beneath Dotson (27.7 Gt a$^{-1}$) in 1996 was higher than beneath Crosson (10.9 Gt a$^{-1}$). This difference is due in part to the much greater area of the shelf (~5200

km$^2$ vs ~2400 km$^2$), but also due to high melt rates near the grounding lines of Kohler and western Smith Glaciers, which feed Dotson.  Notably, sub-shelf melt rates on Dotson are greater than the grounding-line flux for each of the years surveyed, resulting in loss of shelf volume irrespective of its calving rate

Figure 3 also shows melt for different regions of each shelf. The greatest increase in melt occurred on western Smith

Glacier (SW1 in Figure 3), but the most intense melt (~23 m a$^{-1}$) in 1996 was within 10 km of Kohler's grounding line (K1 in Figure 3) and rates there remained high through our study period.  Melt rates ~10km farther downstream from Kohler (K2) were lower (~7 m a$^{-1}$) but doubled during the study period.  The areas of western Smith Glacier with the highest melt (SW1 in Figure 3), found both here and elsewhere (Khazendar et al., 2016), had ungrounded since 1996 (Figure 1).  Melt rates increased through the study period beneath all portions of Crosson except that nearest the

calving front (C2 in Figure 3).  Melt over areas that were floating throughout the study period is unaffected by ungrounding, so the increasing melt in these areas is indicative of a change in ocean forcing through this time.  By contrast, the decrease in melt rates from 2010-2014 may have been caused either by the ice draft shallowing into cooler water or by a change in ocean forcing.

Table 1 compares our estimates of flux and melt to those from previous studies. Our estimates of grounding-line flux agree with previous estimates (Depoorter et al., 2013; Mouginot et al., 2014; Rignot, 2008; Rignot et al., 2013).  Our estimate of the partitioning of the mass loss, however, differs substantially from Rignot et al. (2013) and Depoorter et



al. (2013); this discrepancy stems from different values of the calving-front flux, as well as different data used to determine the thinning. Published estimates of thinning rates range from 2.6 m a$^{-1}$ to 5.6 m a$^{-1}$ over Dotson and do not agree to within their stated errors, leading to a difference of 16 Gt a$^{-1}$ when applied to the whole shelf (Paolo et al., 2015; Pritchard et al., 2012; Shepherd et al., 2004). The thinning rates measured via radar altimetry use a longer

time series of thickness data (Paolo et al., 2015), and so we expect these values to be more representative of the average thinning over our study period than previous estimates (Depoorter et al., 2013; Pritchard et al., 2012; Rignot et al., 2013). We note, however, that the range of published thinning rates leads to much larger errors than those we report propagating only the stated error from Paolo et al. (2015), implying that we likely underestimate the error in our melt and flux calculations. Using different thinning rates, prior studies have rates of loss for ca. 2009 ranging from 36 Gt

a$^{-1}$ (Depoorter et al., 2013; Rignot et al., 2013) up to 63 Gt a$^{-1}$ (Shepherd et al., 2010). While using different thinning rates substantially alters the estimated melt, melt rates on Dotson calculated using any of these values are larger than the grounding-line flux. Thus, using a different thinning rate within the range of published values would not substantially alter our qualitative conclusions, though it would imply greater magnitude of melt.

### 4.2 Buttressing

Previous work has shown rifts along the margins causing gradual separation of Crosson from the seaward portion of its embayment between 1984 and 2012 (Macgregor et al., 2012). Our tracing of more recent ice-front positions shows that this separation has continued to the present (Figure 4a). We find that in 2014 the rifts on the eastern margin of Crosson connected to the ice front, effectively detaching the last ~35 km of the shelf from the right side of the embayment (see labels in Figure 4). Since the ice shelf was already separated from its western margin, this detachment

left the outer ~35 km essentially as a floating ice tongue within its embayment. Since there have been no large calving events, the central portion of the ice front advanced back to near its 1972 position, but provided no buttressing and did not change the force balance.

There was also a progressive loss of contact between the front of Crosson and the Haynes Glacier tongue from 1984-

2004 (Macgregor et al., 2012). These changes continued through 2012, and rifting also increased through this period (Figure 4b and c). We found the linear extent of ice along three transects near the front of Crosson through time to determine the portion of the ice front that may have experienced back force from neighboring ice. This ice extent is similar to the calving-front position for Haynes Glacier found in MacGregor et al. (2012), but we used two additional transects to determine the locality of effects on the front of Crosson, the Haynes tongue, and the Thwaites tongue. We

find a generally decreasing trend starting at least as early as 1984, with reduction to almost no area in contact by 2004 (Figure 4d). The approximately steady decline of this tongue suggests there was no sudden drop in resistance at the margin of Crosson, and any speedup caused by the breakup of this tongue would most likely have been gradual, beginning at least as early as the 1980s and continuing into the 2000s.



### 4.3 Modeled Weakening

Figure 5 shows the inferred enhancement through time from the diagnostic model, which indicates a reduction in the strength of Crosson's margins from 1996 to 2014. By contrast, there is no notable change in the inferred rheology of Dotson. Stiff ice at the calving front of Crosson is similar to results from Thomas and MacAyeal (1982), which they attribute to ice that is thinner and cooler than expected. The general pattern of weakening is consistent with areas of high shear, where increases in strain heating, crystal fabric, or rifting could have caused a positive feedback with speedup; faster ice motion could have caused weakening through strain heating or rifting, allowing further speedup. Because we do not account for spatial variability of thinning within each shelf, what we interpret as weakening could in fact be a result of thinning; i.e. we may infer weaker ice in regions where ice is in fact simply thinner. Similarly, areas where the model identifies strengthening may result from local thickening on those portions of the shelves. While this ambiguity prevents us from interpreting changes in enhancement as a particular physical process, it does not adversely affect our determination of which areas afforded more or less resistance to flow.

### 5 Discussion

Here we examine how variations in ice thickness, melt, and ice strength appear to have affected the flow pattern, grounding line position, and mass balance of these ice shelves. Because of the differing behavior of the shelves, we address different processes over each shelf. On Dotson, melt alone exceeded flux onto the shelf for every year surveyed, implying elevated melt rates at least as early as 1996. Crosson's speedup, which caused more ice to reach the calving front, may have caused its imbalance, and we are unable to determine if Crosson's melt rates would be sustainable in the absence of this speedup. Thus, we first focus on the causes and implications of Dotson's elevated melt and then discuss Crosson's speedup in the following section. Finally, we compare the estimates of melt from this study to previous work and discuss the implications for the dynamics of the system.

### 5.1 Causes of Dotson's imbalance

Changes in the Amundsen Sea Embayment have generally been attributed to oceanic forcing (Joughin et al., 2010; e.g. Shepherd et al., 2004). It is possible that an increase in ocean heat content prior to our study period may have contributed to Dotson's observed imbalance. In this case, elevated melt rates on Dotson may have directly led to a retreat of its grounding line, resulting in further exposure to melt and continued retreat. Alternatively, some of the change may have resulted from ongoing thinning. For example, model results indicate a glacier catchment may thin for decades, leading to an abrupt retreat of the grounding line (Jamieson et al., 2012; Joughin et al., 2014). Thus, even modest but sustained thinning could perturb the grounding line from a stable position. In either case, once perturbed, the grounding line may undergo a rapid retreat to another stable position, thus leading to greater exposure of sub-shelf area and elevated melt (e.g. Jenkins et al., 2016). We now present evidence that suggests a large retreat of Kohler's grounding line took place prior to our study period, then evaluate whether ongoing thinning or ocean forcing caused this grounding-line retreat and Dotson's mass imbalance.





### 5.1.1 Prior retreat of Dotson's grounding line

The bathymetry near Dotson's present-day grounding line suggests that any recent unstable retreat would likely have taken place at Kohler Glacier. While the Smith Glacier grounding line was positioned on a bedrock high in 1996, Kohler Glacier's grounding line was ~1200m deep, just upstream of a ~20 km long retrograde slope (Figure 6b) (Millan et al., 2017). If the grounding line formerly was positioned downstream of this retrograde slope (e.g. dashed line in Figure 6b), it likely would have retreated rapidly if perturbed. The unsustainably high melt rates in this area (K1 and K2 in Figure 3) indicate that such a retreat of Kohler's grounding line had likely taken place recently prior to 1992. The overall mass balance for Dotson implies that exposure of K1 and K2 to ocean forcing would have been the difference between sustainable and non-sustainable melt. Comparing the calving flux and the melt from D1 and D2 alone to the steady-state grounding-line flux (Figure 3) shows that Dotson would have been approximately in balance with 1996 melt rates if only this area had been exposed. Adding in the melt found beneath K1 and K2, even at 1996 rates (the lowest in our study period), however, would have resulted in melt exceeding grounding-line flux. Extrapolating back using current thinning rates of ~2.5 m a$^{-1}$, most of the shelf would have been entirely grounded 150 years prior, implying that the melt rates within this overdeepening could not have been sustained over centennial timescales. Thus, it is possible that some time in the several decades prior to 1990s, the grounding line retreated several 10s of kilometers from our hypothesized position (Figure 6 dashed line), increasing exposure to melting in the K1 and K2 region (Figure 3).

Velocity data provide an additional constraint on the possible timing of a retreat of Kohler's grounding line. Feature tracking of Landsat imagery gives sparse velocity measurements on Dotson for the period 1974-1982, and comparison of these data with recent velocities suggests that there was no substantial speedup of Dotson from 1974 to the present (Lucchitta et al., 1994; Rignot, 2008). Speedup near the grounding line usually accompanies ungrounding due to the associated loss of basal resistance, so the lack of speedup suggests any substantial grounding line retreat on Dotson likely took place prior to 1974. Combining this lower limit on retreat initiation with the upper limit placed by the unsustainably high melt rates, we infer that the imbalance began years to decades before 1974. Previous work has found that grounding-line flux over Crosson and Dotson combined approximately balanced accumulation in 1974 (Rignot, 2008). That balance is inferred primarily from speedup of the outer portion of Crosson during that time; however, speedup on the outer portion of the shelf can take place independently of speedup near the grounding line (see section 5.2 below). On the other hand, the near-constant velocities on Dotson from 1974-1996, combined with the imbalance in 1996, suggest that Dotson may have been out of balance in the 1970s. Moreover, even if Dotson's grounding-line flux were in balance with upstream accumulation, the melt rates on the shelf may have been elevated. We therefore consider the available data to be ambiguous about the state of balance from the mid 1970s to early 1990s, and thus reconcilable with this proposed timing of retreat initiation.

In summary, observations of grounding line position and velocity are consistent with past retreat of Kohler Glacier due to marine ice-sheet instability (e.g. retreat from the example position shown in Figure 6b), and this retreat likely occurred in the years to decades before 1974. This consistency does not, however, identify the initial cause of that





retreat, which could have been triggered either by an unstable response to ongoing thinning or by an increase in basal melt. To further investigate these two scenarios, we discuss how the present-day geometry of Dotson, particularly a large basal channel (Figure 1c and 6a), informs our understanding of its history of melt and flow.

### 5.1.2 History of melt

Surface features on an ice shelf can preserve information about the flow and melt history as ice advects seaward (Fahnestock et al., 2000). To help differentiate whether ongoing thinning or ocean forcing triggered Kohler's grounding line retreat and Dotson's imbalance, we use the large surface trough visible in the surface topography (Figures 1c and 6a) to estimate the onset and spatial extent of changes on Dotson. Present day melt rates in this channel, particularly after it turns towards the ice front, have been previously studied (Gourmelen et al., 2017), but

here we focus on the portion of the channel that is transverse to flow. The surface expression of this portion of the channel is 5-10 km wide in a region where ice speeds are 150-200 m a$^{-1}$. Downstream of this channel, the shelf thickens by ~200 m. In steady state, an ice shelf should only thicken along flow if there is basal freeze-on, high SMB, or a narrowing of the embayment. Dotson has nearly parallel sidewalls and the magnitude of the downstream increase in thickness is large compared to the SMB, so these causes can be eliminated. We also eliminated the possibility of

freeze-on being the primary cause of the channel by comparing radar observations to floatation levels. Where marine ice is present, ice penetrating radar generally does not propagate into the saline marine ice layer, hence the radar only records the thickness of the meteoric ice. As result, significant differences between the radar (meteoric-only) and hydrostatic-equilibrium (full column) derived thicknesses indicate the presence of marine ice (Crabtree and Doake, 1986; Robin et al., 1983; Thyssen, 1988). No such differences are found for Dotson, indicating little or no marine ice

(Further details can be found in the supplementary materials). Thus, having ruled out other causes, we conclude the channel was produced by a transient change in ice flux or melt.

The channel originates near the grounding line on the eastern (shelf-right) margin of Dotson and continues along the grounding line to the western (shelf-left) margin and extends to the shelf front. This pattern is similar to the pattern of

ocean circulation inferred from oceanographic measurements. Available data show warm inflow at the eastern margin and meltwater-laden outflow at the western margin, indicating clockwise circulation beneath the shelf (Ha et al., 2014; Miles et al., 2015). This circulation pattern suggests that the channel originates where the warm CDW inflow first comes into contact with the ice shelf draft. Generally, melt that begins at depth is sustained by entrainment of additional warm water as the buoyant meltwater plume rises along the underside of the ice (Jenkins, 2011). Basal

melt channels have been observed in many locations around Antarctica (e.g. Alley et al., 2016), and are thought to form in the location where the meltwater plume rises along the bottom of the shelf (Marsh et al., 2016; Stanton et al., 2013). The shallowing of this channel as it curves along the grounding line to the western margin of Dotson is consistent with this model of channel formation. The abrupt beginning of the channel on the eastern margin and the continuation of the channel to the shelf front on the left are thus consistent with the available oceanographic

constraints. However, high resolution measurements of melt from 2010-2016 do not show high melt rates throughout this channel, but rather localized melt near the grounding line along the western portion of the channel before it turns



toward the shelf front (Gourmelen et al., 2017). This present pattern of melt suggests transverse portion of the channel is at least partially a mark of past thinning rather than a signal of currently elevated melt.

To constrain the timing of the formation of the channel, we assumed that it resulted from a perturbation at the grounding line and used the 1996 velocity to determine how long the perturbation would take to advect to the current channel location (Figure 6a). For this calculation, we assume the perturbation happens at or close to the grounding line; since grounding line positions and surface elevations before 1992 are unknown, this is the simplest assumption. Additionally, the present-day concentration of melt immediately near the grounding line in the western portion of this channel suggests such an assumption is reasonable (Gourmelen et al., 2017). Estimates for the timing of the perturbation are 30-45 years before 2013 (i.e. 1968-1983) for various points along the channel. This timing is consistent with our inference of when a retreat of Kohler's grounding line may have occurred, suggesting that there were widespread changes in melt near Dotson's grounding line in the decades prior to 1974, and that the channel and the retreat of Kohler's grounding line were both consequences of these changes. Grounding-retreat is thought to have begun on other glaciers in the region at a similar time (e.g. Jenkins et al., 2010), but the complex nature of ice streams' response to perturbations prevents interpreting a synchronous change as evidence of synchronous forcing.

While ocean forcing is likely responsible for these changes, we cannot fully eliminate other possibilities. The most likely alternative is that retreat of Kohler Glacier's grounding line, perhaps due to ongoing thinning, resulted in changes to the circulation patterns beneath the shelf, causing increased melt and channel incision. Exposure of deeper ice at Kohler's retreated grounding line could have led to increased melt not only at the grounding line, but also downstream as meltwater from the grounding line entrained warm water. This increased outflow, which would have risen along the bottom of the shelf as it flowed northward along the left margin of Dotson, may have enhanced overall circulation beneath the shelf, incising the channel as additional warm water cycled through the cavity. Such grounding line retreat may have also resulted in speedup and thus dynamic thinning in the area of the channel as the portion of shelf downstream of Kohler thinned and resistance to flow was reduced. Incision of the transverse portion of the channel through dynamic thinning would also be consistent with the low melt rates in that area found by Gourmelen et al. (2017).

### 5.2 Causes of Crosson's speedup

Understanding the changes in flow in this area, particularly the observed changes in Crosson's velocity, is important for predicting the future stability of this system as well as understanding the causes of increased calving and grounding-line flux through the study period. Crosson's speedup was strongest in two distinct regions: near eastern Smith Glacier's grounding line and near the eastern (shelf-right) margin of the calving front (Figure 2). These two regions suggest that multiple processes may have influenced the speedup, and so we discuss these regions separately.



### 5.2.1 Speedup of the outer shelf

The most notable changes near Crosson's calving front are the loss of the Haynes Glacier tongue and the increased rifting near the eastern margin. The rifts in this area coincide with inferred enhancement from our diagnostic model (Figure 5), suggesting that the strength of this area is important to the flow of the shelf. To understand how this rifting

may have affected the shelf, we compared the enhancement to Crosson's compressive arch (contour in Figure 5).

The spreading side walls of Crosson's embayment result in a transition from compression to extension that is nearly perpendicular to flow, with most changes to enhancement downstream of this transition (Figure 5). The overall dynamics of Crosson would have been relatively insensitive to weakening seaward of this transition, but changes

landward of this transition would have resulted in broad speedup. Because the weakening on Crosson was primarily just in the tensile area, the effects of the weakening would have been isolated to the freely spreading area downstream of the arch, which is consistent with the observed speedup (Figure 2b&c). This weakening is unlikely to have had a widespread effect on the shelf dynamics, but breakup of the outer shelf exposes the inner shelf to further weakening. This weakening would involve dynamic effects beyond the immediate vicinity of the rifting/damage, possibly causing

speedup upstream on the portions of Crosson that remained at nearly constant velocity through the 1990s and 2000s. By contrast, there has been little weakening of Dotson.

The local nature of the speedup on the outer portion of Crosson suggests a local cause (e.g. the breakup of neighboring ice). Loss of resistance at the shelf front can increase calving rates, reducing lateral contact area and initiating speedup

(Cassotto et al., 2015). The Haynes Glacier tongue may have provided a small but critical shear resistance on the corner of the shelf, and the gradual loss of this tongue may have allowed a cycle of increasing speedup and weakening of Crosson's eastern corner. The speedup of this area of the shelf began before the 1990s (Lucchitta et al., 1994; Rignot, 2008) and continued through 2009, and the Haynes tongue broke up throughout this same period. However, due to feedbacks between strain heating or rifting and speedup, the observed Crosson weakening may either have

caused or been caused by its speedup, and the simultaneous breakup of the Haynes tongue may have been unrelated or a response to the same forcing that weakened Crosson. Utilizing a parameterization of damage (e.g. Borstad et al., 2016) in a prognostic model, which would let weakening of the shelf evolve with speedup, could help to identify whether these changes initiated with the breakup of the Haynes tongue or some other forcing.

### 5.2.2 Speedup near the grounding line

Speedup near eastern Smith Glacier's grounding line is most likely associated with loss of basal resistance caused by grounding-line retreat. The grounding line could have been perturbed directly through increased basal melt, either synchronously or asynchronously with the increased melt on Dotson. However, the changes in Crosson's grounding-line position also could be a result of the ungrounding on Dotson. Thinning of western Smith Glacier would have led to thinning of the eastern branch as well. Such thinning could have led to ungrounding of the trunk of eastern Smith

Glacier and thus loss of basal resistance, dynamic thinning, and speedup. The velocity and thinning signatures of





increased melt at eastern Smith Glacier's grounding line and of a retreat caused by thinning propagating over from Dotson could be addressed with a prognostic ice-flow model.

### 5.3 Comparison to previous flux and melt estimates

Our melt estimates for Crosson and Dosson are lower than the peak values reported by *Khazendar et al.* (2016), which
is expected because our methods measure a fundamentally different quantity. While we measure total melt rates over large regions, Khazendar et al. (2016) used RES to estimate ice-bottom elevation change at particular points, and assumed that these thickness changes directly represent basal melt. The total melt rate at each point is higher than the thinning rate they calculate, since some amount of melt would still occur if the shelf were in steady state. The estimation of anomalous melt also requires an assumption that dynamic thinning is negligible; Khazendar et al. (2016)
justify this assumption based on observed thinning rates for grounded ice, which are low compared to their peak melt estimates. However, the rates of flux divergence on the shelves are comparable to the thinning rates in many areas. We calculated flux divergence using 1000-m velocity grids for the period 2009-2014 and applied this correction to the radar observations, resulting in our estimate of anomalous melt rate differing by over 100 m a$^{-1}$ in some places (further detail can be found in Supplementary Materials). Our recalculated values for RES-derived melt rates range
up to 200 m a$^{-1}$ (which is up to 70 m a$^{-1}$ higher than peak rates in Khazendar et al.), and show a different pattern of melt from that found in Khazendar et al. (Supplementary Table 1 and Supplementary Figure 1). However, the use of repeat thickness measurements to find anomalous melt may still be biased by crevasses or keels advecting through the radar-track intersections, causing incorrect interpretation of structural anomalies as melt. While the polygon-based method we employed is more limited in spatial resolution, it is much less sensitive to these small-scale variations and
thus more representative of melt rates for large sections of the shelves, though we may smooth out areas of peak melt. The method used by Gourmelen et al. (2017) resolves these small scale variations by accounting for advection, but it requires spatially dense satellite coverage and so is temporally limited compared to the methods employed above. All of these methods are highly sensitive to the thinning rate, so the wide range of published thinning rates implies uncertainties are likely underestimated.

### 6   Summary

We used observations of elevation and velocity along with an inverse model to investigate the causes of grounding-line retreat and speedup on Crosson and Dotson ice shelves. These two ice shelves exhibited contrasting responses to changes in forcing, despite similar grounding-line flux and similar changes in melt. Confirming earlier results (Mouginot et al., 2014; Rignot, 2008), we find that both ice shelves were out of balance at the beginning of our
observational period in 1996. Thinning and speedup early in the study period were associated with an ongoing response to earlier changes. Basal melt rates increased on areas that were floating throughout the study period, and total basal melt was further increased as ungrounding exposed more area to melt. The melt rates, grounding line position, and incised channel geometry on Dotson suggest that it began to retreat in the early 1970s or before. A change in ocean forcing years or decades before 1974 likely led to in Dotson's imbalance in 1996. Dotson's
grounding-line retreat and thinning exposed more sub-shelf area, increasing its sensitivity to ocean forcing and likely



contributing to the high melt rates observed in the 2000s. Crosson sped up through our study period, primarily near eastern Smith Glacier's grounding line and near the former tongue of Haynes glacier. This speedup was likely the result of multiple factors, including weakening at its eastern margin and a retreat of eastern Smith Glacier's grounding line. Prognostic modeling of this system beginning in 1996 or before (i.e. "hindcasting"), forced with observed melt
rates, could help test how different initial perturbations to the system would have affected its flow speed and mass balance.

**Acknowledgements.** This work was supported by NASA. D. Lilien and D. Shean were supported through the NASA Earth and Space Science Fellowship (NESSF) Program (NNX15AN53H and NNX12AN36H). I. Joughin was
supported by NNX17AG54G, and B. Smith by NNX13AP96G. Computing resources for ice-flow modeling and for WorldView/GeoEye DEM processing were provided by the NASA High-End Computing (HEC) Program through the NASA Advanced Supercomputing (NAS) Division at Ames Research Center. RACMO2.3 SMB data were provided by J.M. van Wessem (Utrecht University). F. Paolo (JPL) provided thinning rates over the ice shelves.

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





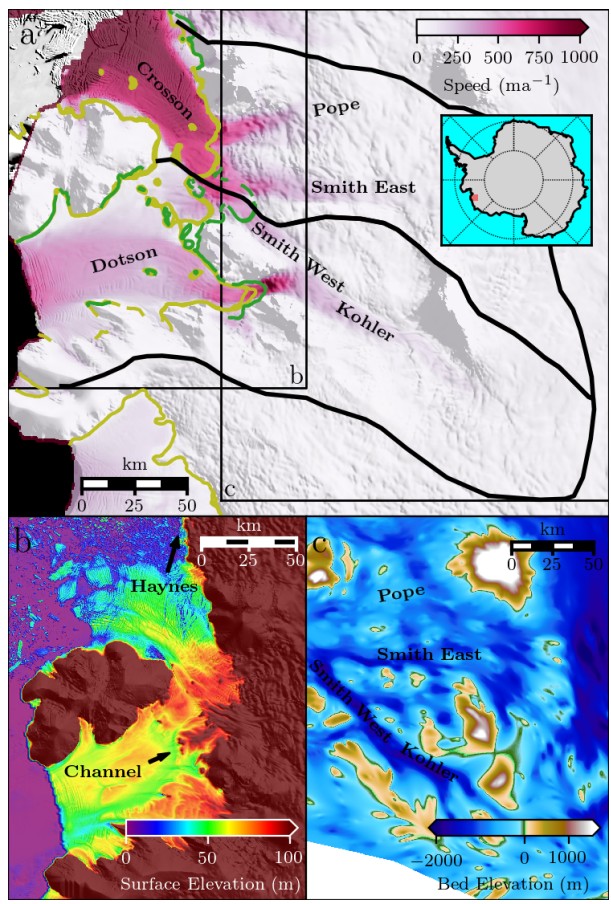

**Figure 1: Overview of study area a) 1996 surface speed overlaid on the mosaic of Antarctica (MOA) (Haran et al., 2013). Yellow and green lines show grounding line positions in 1996 and 2011 respectively (Rignot et al., 2014). Black lines indicate catchment boundaries of Crosson and Dotson used for flux calculations. b) Surface elevation relative to the EGM2008 geoid from WorldView/GeoEye stereo DEM mosaic (Shean et al., 2016). c) Ice bottom elevation relative to the EGM2008 geoid, which represents bed elevation over grounded ice.**



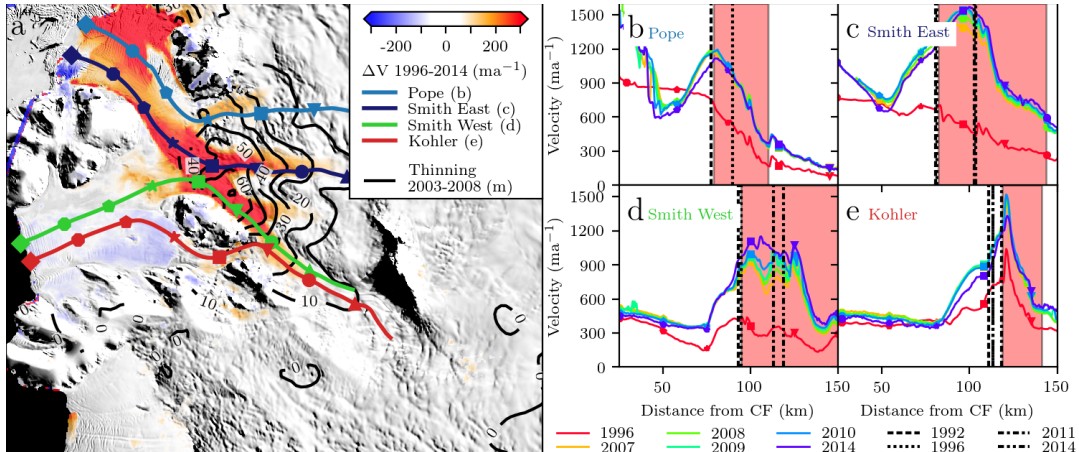

**Figure 2: Recent changes in velocity and surface elevation. a) Color shows change in speed from 1996 to 2014. Contours indicate thinning of grounded ice from 2003-2008 at 10-m intervals, derived from OIB altimetry, ICESat-1 and WorldView/GeoEye DEMs. Colored lines indicate flowlines plotted in panels (b - e). Background is MOA. b – e) Velocity profiles as distance from the 2014 calving front (CF). Dashed lines indicate grounding line positions in different years. Red shading indicates areas that thinned by more than 5 m a⁻¹ from 2003-2008.**





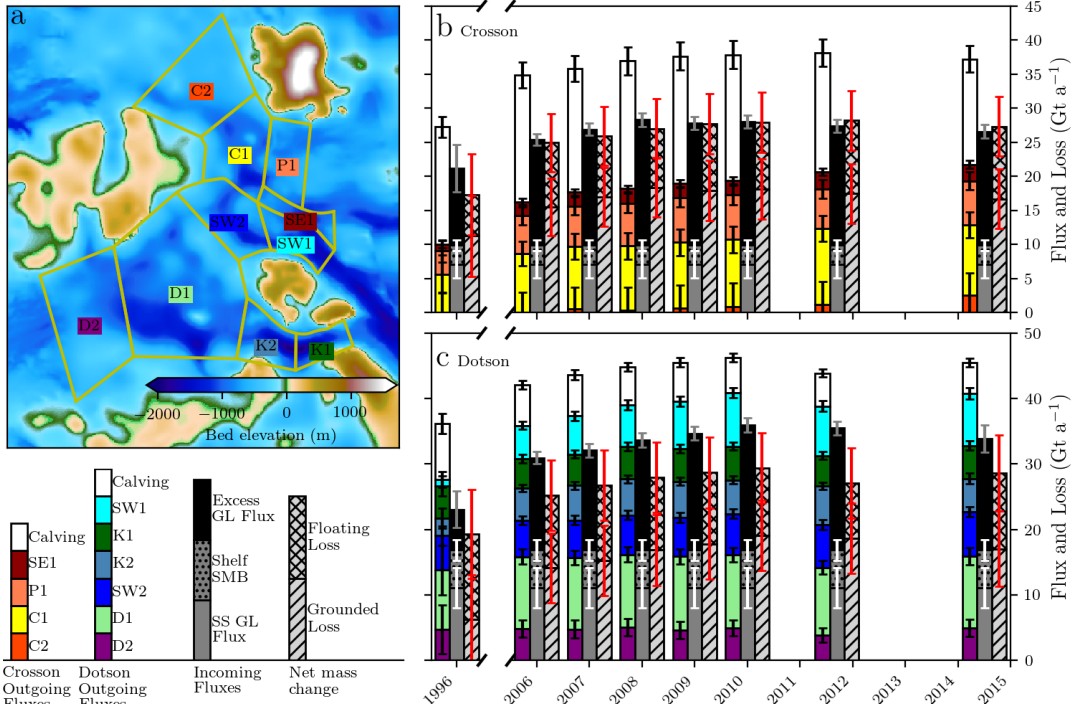

**Figure 3: Flux and melt changes over Crosson and Dotson. a)** Areas used for flux divergence calculations, outlined in yellow, plotted over bathymetry from Millan et al., (2017). **b) and c)** Flux and melt over Crosson and Dotson respectively. Stacked, colored bars show melt over each of the polygons from (a) while white bars on top show calving flux; these bars sum to the total outgoing flux. Solid gray bars show the steady-state grounding line flux (calculated from catchment-wide SMB), the dotted gray bars show the SMB over the shelves, and the solid black bar indicates the additional flux crossing the grounding line each year due to dynamic imbalance; these bars total to all incoming flux to the shelves. Light gray slashed bars indicate the loss of grounded ice (equal to the black bars in the previous column) and the cross-hatched light grey region shows the additional loss of grounded ice (equal to the difference between the first two columns); these total to the annual mass loss rates.





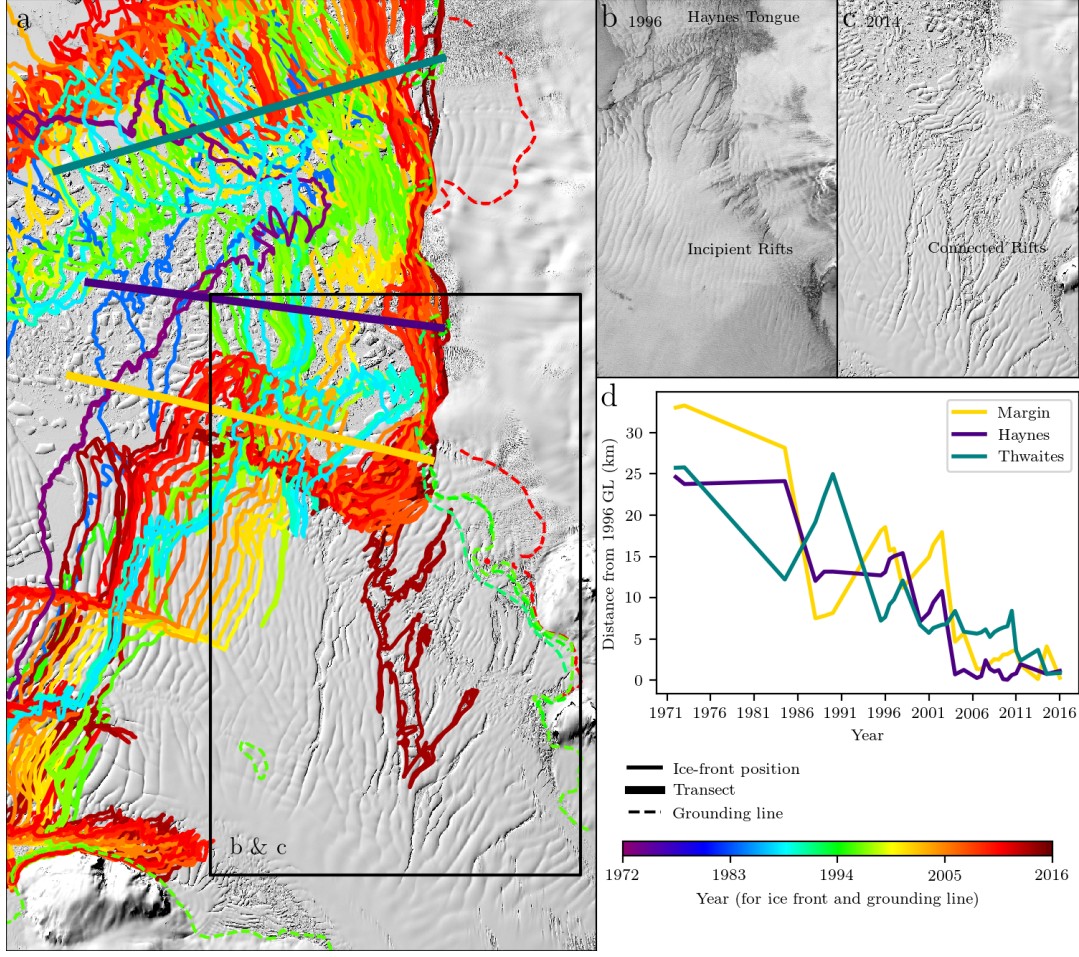

**Figure 4: Crosson Ice Shelf front position. a) Selection of digitized shelf-front positions from MacGregor et al. (2012) and from our work, with grounding lines from Rignot et al. (2014). Teal, purple, and gold overlays correspond to transects used for (d). Box shows the outline of zoomed in area for (b) and (c). b-c) Zoomed in Landsat imagery from 1996 and 2014 respectively. Isolated rifts in 1996 appear connected by 2014, effectively detaching Crosson from its eastern margin. The disintegration of the Haynes Glacier Tongue also occurred during this period. d) Time series of ice-shelf-front distance relative to the 1996 grounding line.**





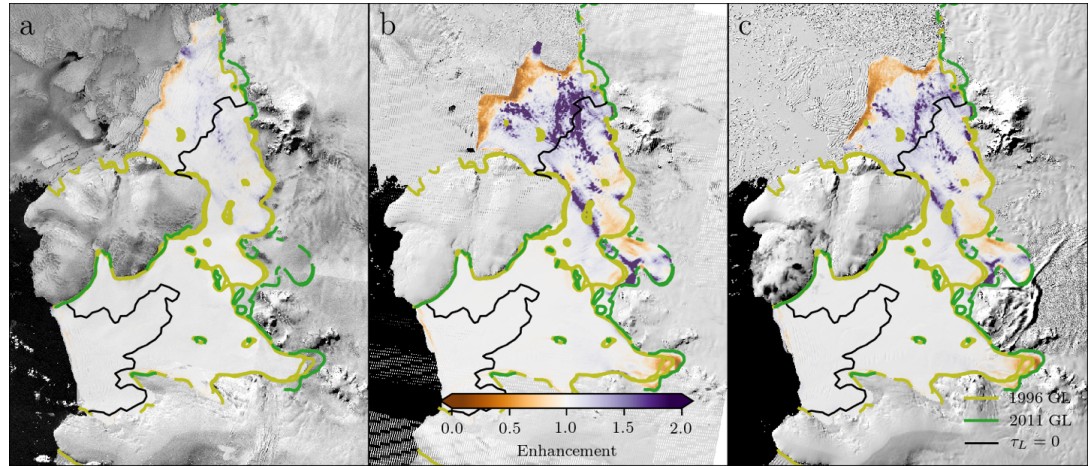

**Figure 5: Inferred enhancement to ice flow in a) 1996, b) 2010, and c) 2014. The enhancement factor gives the depth-averaged weakening of the ice relative to that predicted by a temperature model, with values larger than 1.0 indicating weakening. Black line indicates the zero contour of $\tau_L$ in 1996, i.e. the boundary between the compressive (upstream) and extensional (seaward) regimes. Backgrounds are Landsat images from the same year as the inversion.**



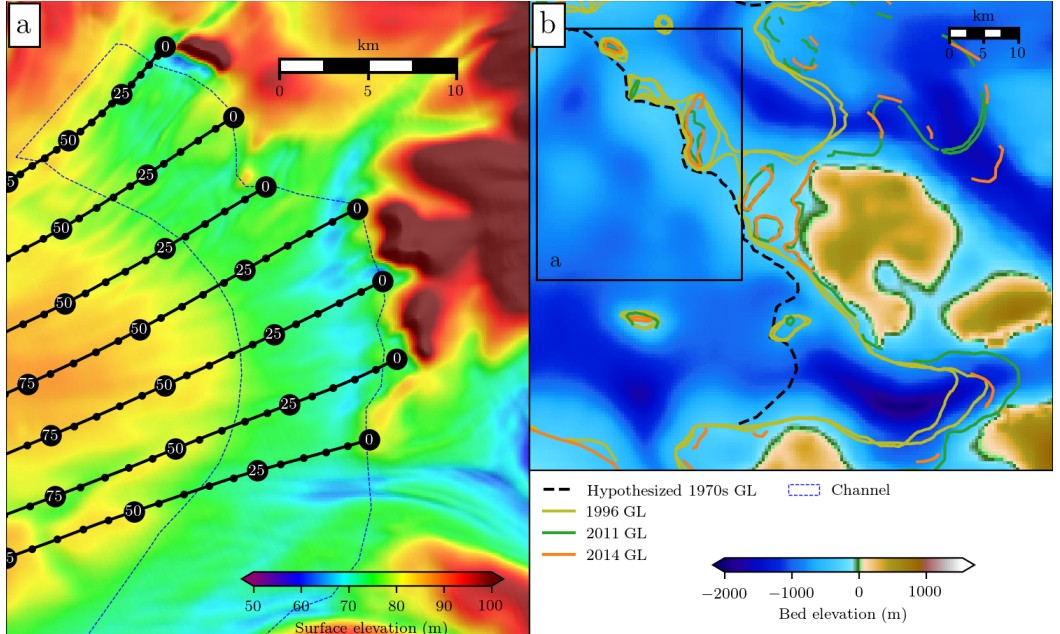

**Figure 6: a) Detail of Dotson surface trough (region outlined in blue). Solid black lines show flowlines calculated from the 1996 velocity, with dots placed every 5 years. Large black circles are spaced every 25 years and labeled with age in years. Background is 2012 surface elevation. b) A possible location of the grounding line before recent retreat (black dashed line). This position is illustrative only, but shows a location that would have been susceptible to a retreat that would have increased melt rates and grounding line flux. Other lines show grounding line positions as above. Black box shows location of panel (a). Background is bathymetry (Millan et al., 2017).**



| | Grounding-Line Flux | | | Calving Flux | | | Basal Melt Rate | | | Volume Change | | |
|---|---|---|---|---|---|---|---|---|---|---|---|---|
| | | (R13) | (D13) | | (R13) | (D13) | | (R13) | (D13) | | (R13) | (D13) |
| Crosson | 25±1 | 27.4±2 | -- | 18±2 | 12±2 | -- | 19±3 | 38.5±3 | -- | -9±2 | -19±1 | -- |
| Dotson | 30±1 | 28.4±3 | -- | 5±1 | 6±1 | -- | 41±2 | 45.2±3 | -- | -11±3 | -17±2 | -- |
| Total | 55±2 | 55±5 | 51±5 | 25±3 | 17±3 | 18±2 | 60±5 | 83.7±6 | 78±7 | -20±4 | -36±3 | -36±3 |

**Table 1: Comparison of fluxes and melt between this study (white columns) and those from *Rignot et al.* (2013) (R13) and *Depoorter et al.* (2013) (D13). Values from this study are for 2010, while the *Depoorter et. al.* values are for 2009, and the *Rignot et al.* values are for 2007-2008. Columns show grounding-line flux, calving flux, basal melt rate, and volume change. All values are given in gigatons per year (Gt a$^{-1}$).**

