# Peer review of "Changes in flow of Crosson and Dotson Ice Shelves, West Antarctica in response to elevated melt"

_The Cryosphere, 2017_

## Referee Comment (RC1) · Anonymous Referee #1 · 7 Feb 2018

This paper describes observations of mass budget for the Crosson and Dotson ice shelves in the Amundsen Sea sector of the Antarctic ice Sheet and attempt to provide a narrative for the timing of imbalance and for the succession of events and interaction that led to the current state of imbalance and mass loss of both floating and grounded ice in the region. The paper integrates new and existing observations with modelling in a balanced way in the context of existing literature. The manuscript is well structured, detailed and illustrated. I support the publishing of the manuscript with only minor changes.

General comments:

[Figure]

I find the title vague and confusing; how varied, what response, and how varied are the elevated melt in the first place. Do you imply here that the same, elevated, melt under both ice shelf induce the various response? From the findings of this paper it does not look to me that this is the case.

Thinning rates used to calculated ice shelf elevation back in time are resolved over spatial resolution of 10's of kilometres; recent work have shown that, at least for Dotson, thinning is highly localised to the grounding lines and on the western margin of the shelf. You should at least address the impact that this might have, or not, on your analysis. You allude to it in section 4.3 but can you be more specific as to where this might be the case?

Specific comments

Page1

L12: Ambiguous, suggests rephrasing. Is what is meant here that the melt under Dotson is dominating the mass loss? Or that the melt under Dotson is driving the mass loss elsewhere? From fig.3 it looks as though the mass loss of Dotson and Crosson are pretty much equal, that the increase in melt rate through he study period is similar, and that one is dominated by loss through basal melt (Dotson) and the other through calving (Crosson).

L17: The preceding line gives an explanation for the observation, but this one does not - what is causing the lack of response of Dotson's velocity?

L36: The terminology is indeed varied - Smith West is sharing a drainage basin and an ice shelf with Kholer, shouldn't this then be Kholer East instead as in Mouginot et al., 2014?

L15: Ambiguous, suggest: "No similar changes in extent were observed over Dotson"

L23: "over" -> "under"

L24: "much higher than the surrounding ice" with the exception of the grounding line regions of Smith and Kholer

L15: Medley has no sup. Material.

L27: What air thickness is found here and how does this compares with modelled values? There are evidence that firn-air content found using this approach differs from model output, so well worth addressing this here. A map in Supp. Mat. For example.

Line 6: The longer dataset also agrees with recent, high-resolution, estimates. Line 10: Reference Shepherd 2010 is missing

L2: You might want to compare and contrast the weakening described here with the shelf weakening and impact on speedup of Smith described in goldberg et al (2016, fig 3) - I think it is relevant here as there are similitudes but also new patterns, especially along the Eastern margins.

L14: There is also no indication of freeze-on from EO-based basal melt extraction

L35: There seems to be melting occurring at the southern end of this portion of the channel. In fact it looks to me that the effect of the advection is visible along most of the length of the channel (ice flow is transverse to the channel up to about half-way down the shelf) as the locus of melt occupies only a fraction of the entire topographic depression. Then, can other sections of the channel be used to derive timing? How

would ice divergence influence the time-evolution of this channel's width and depth (e.g. Drews et al., 2016)? How does this timing relate to timings deduced from grounded ice thinning and ice shelf channel formation (e.g. Konrad et al., 2017; Gourmelen et al., 2017)?

Line 21: Ok, how does these melt estimates fit in your comparison earlier in this paragraph between melt obtain here and melt rates obtained by Khazendar et al?

Fig. 3: b and c should have the same y-axis range (e.g. [0 50]) caption: steady state (SS) The cross-hatched light grey bar is labelled floating ice in the figure and grounded ice in the caption - should it be floating ice everywhere?

---

## Referee Comment (RC2) · Anonymous Referee #2 · 9 Feb 2018

This manuscript presents a detailed analysis of the changes in speed and grounding-line position of two rapidly changing ice shelves, and their respective catchment basins, in the Amundsen Sea sector: Crosson and Dotson. The work combines a diverse set of measurements: satellite remote sensing data of horizontal velocities and elevation changes, airborne radar sounding for ice thickness, satellite imagery to determine ice-shelf front position, and published estimates of ice-shelf thinning rates. The authors then estimate the ice viscosity using a numerical model and perform an analysis of changes in ice-shelf flux, basal melt, and calving rates. Their results show that Dotson and Crosson ice shelves exhibit different dynamic responses to similar perturbations, and that is likely that both ice shelves were out of balance well before the beginning of

the satellite record (the 1990s).

General comments:

The manuscript presents a comprehensive study on the mass balance and velocity state of Dotson and Crosson ice shelves. The work provides some important insights into the origin of the current observed mass imbalance on these ice shelves and their respective catchment basins.

I couldn't find major issues with the work. Overall, the manuscript is well written, and the science is solid. I have, however, a few comments/suggestions that I think need further clarification and will improve the presentation of the work. These edits should be straightforward to incorporate.

In particular, the Summary section needs a bit more work. I think there should be a preference for the active voice in the 'Summary' (e.g. we analyzed, we found, we concluded, we showed), briefly stating the implications of your findings, as well as summarizing the path that leads to your conclusions. I feel this is missing here.

On the melt rate calculation; Fig. S1 shows melt rates estimates for 1-5 year time intervals. Since these are anomalies in basal melting, it is expected (in fact it has been demonstrated, e.g., Jacobs et al., 2013; Dutrieux et al., 2014; Christianson et al., 2016; Paolo et al., 2018) that melt rates are (highly?) sensitive to changes in ocean forcing in the Amundsen Sea sector, which fluctuates substantially at interannual timescales. So, I am unsure how much "weight" one can put on these (highly variable) short-term estimates of basal melt rates in the context of past (longer-term) dynamics of the ice shelf.

I am a bit confused about the fact that you used an average thinning rate for the entire ice shelf, which it might be fine for your polygon estimates, but then compare point estimates of basal melt against Khazendar et al. (2016). How meaningful is this comparison? If I understand correctly, the spatial variation in your melt-rate estimate is

determined mostly by the flux divergence since you do not account for the spatial vari-ability in thinning rates, is that right? If so, how significant do you expect the changes in dH/dt across the ice shelf to be compared with the spatial changes in horizontal velocity?

The title needs to be a bit more informative (Observed elevated melt? Over what period? Elevated relative to what? What kind of response?).

To facilitate the reviewing process, please use continuous line numbering throughout the entire manuscript, and number every single line.

Specific comments:

Lines btw 5-10: "remotely sensed datasets". Which datasets?

Lines btw 10-15: "melt" => "elevated melt" or "melt in excess" (there is basal melt in the steady state)

Lines btw 20-25: "instability" => "internal instability"?

Lines btw 20-25: "are the dominant source of sea level rise" => "are currently the dominant source of sea level rise" or "are the dominant source of current sea level rise"

Lines btw 20-25: "ice flux on ice thickness" => "ice flux across the grounding line on ice thickness there"

Lines btw 25-30: "deeper ocean waters are generally warmer in these systems". Care-ful here, this cannot be a general statement. In locations where CDW is present (not everywhere), there is more of this warm water reaching the deep grounding lines be-cause the CDW is denser than surface waters. This is different from saying "deeper

waters are warmer".

Lines btw 30-35: "recent speed and thickness change" => "recently observed speed and thickness changes"

Lines btw 10-15: "significant changes in the extent of Crosson". How significant? Can you provide a percentage of area change?

Lines btw 30-35: "not straightforward" => "challenging"

Lines btw 30-: How is the uncertainty propagated for the values predicted by the quadratic function where there is no data at all? This uncertainty should increase (substantially) as the predictions get further away from the period constrained by data.

Lines btw 30-: Not sure I understand the following statement: "The results from this method more closely match the available ICESat-1 data than using the elevation and thinning rate from 2004 alone, as was done in Mouginot et al. (2014)". In what sense it matches better ICESat (2003-2009), which is outside of the extrapolated range (1996-2002)?

Lines btw 5-10: "The thinning rates on the 10 grounded ice are more accurate since they lack tidal effects". This is not the main reason why altimetry thinning-rate estimates may be more accurate over grounded ice. Over (rapidly-changing) grounded ice the signal-to-noise ratio is much higher than over floating ice where the altimeter only "see" about 10% of the thickness-change signal due to hydrostatic balance. Also, tide models are reasonably good (outside of the grounding zone) at the current stage.

Lines btw 15-20: "This method reduces many of the artifacts that can occur when inter-polating sparse ice thickness measurements while avoiding making any assumptions

about the present state of balance (e.g. assumptions for mass conservation methods (Morlighem et al., 2011))". It is still interpolation, right? Meaning that regardless of any "smart" weighting, you don't have information in between the sparse thickness samples. The mass conservation approach uses additional information (velocities) to fill in those gaps. Bottom line, one interpolation method, another interpolation method... it is still interpolation.

Line 1: "m_s is the basal melt rate". m_s => m_b

Line 5: I think a more appropriate reference for RACMO2.3 SMB is "Van Wessem et al. (2014)".

Line 16-17: "but this choice should not have affected the overall spatial pattern". How sure can you be about this (i.e. what is the effect of many small artifacts on the overall result)?

Line 9: "and any additional amount entering the shelves in each year". Suggest adding "by precipitation over the ice shelves" or "through ice-shelf surface accumulation".

Lines 12-13: "would not substantially alter our qualitative conclusions". You can remove "qualitative" here.

Lines 8-9: "implying that we likely underestimate the error in our melt and flux calculations". Not necessarily. The discrepancy in the thinning rates on those studies is mostly due to some using a 5-year (laser) altimetry record while the other using an 18-year (radar) altimetry record. So, the different thinning rates are likely representing different timescales. For example, as shown more recently by Paolo et al. (2018), there are

large fluctuations in ice-shelf thickness at ENSO timescales (∼4-5 years), a time span comparable to the ICESat period.

Line 5: "ice that is thinner and cooler than expected" => ???

Lines 13-14: "the downstream increase in thickness is large compared to the SMB". Can you provide the typical magnitude of SMB over this region (e.g. from RACMO) to put things in context?

Lines 16-24: When estimating basal melt rates using ice-shelf surface height changes (e.g. estimates derived from altimetry), there is a substantial uncertainty associated with fluctuations in surface mass balance, which can have large amplitudes at seasonal-to-interannual timescales. Moreover, it has been shown that, often, increase basal melting and surface height change are out of phase (i.e. they can go in opposite directions) (e.g. Paolo et al., 2018).

Line 28: "similar" => comparable

Line 28: "changes in melt" => changes in basal melt

Line 30: "were associated" => by whom?

Line 32: "total basal melt was further increased" => by whom?

Please avoid the passive voice. It is not clear whether you are stating known facts (from the literature) or stating your conclusions.

Line 34: "A change in ocean forcing years or decades before 1974 likely led to in Dotson's imbalance in 1996". This is highly speculative, particularly regarding "a change in ocean forcing". It is OK to speculate but clarify (in the Summary) what led you to this

conclusion.

Page 17:

Lines 4-6: Are you suggesting this as future work? If so, please clarify you are pointing the future direction of work needed; and justify why we would need such direction (i.e. what's the relevance in the context of understanding and predicting future ice-shelf/ice-sheet loss).

Supp. Page 2:

Line 18: "and we are unsure of the exact method used by Khazendar et al". Why don't you ask them?
* * *

---

## Author Comment (AC1) · 9 Mar 2018

While addressing the reviewer's concerns, we have been unable to fully understand the following comment:

"Page 13 L14: There is also no indication of freeze-on from EO-based basal melt extraction"

In particular, we are unfamiliar with the abbreviation EO in this context. If revision of this line is necessary, could the reviewer please provide a reference or further clarification about this comment?

---

## Referee Comment (RC3) · Anonymous Referee #1 · 14 Mar 2018

By EO I mean Earth Observation. Published basal melt rates based primarily on EO data (e.g. Rignot et al., 2013; Depoorter et al., 2013; Gourmelen et al., 2017) show no evidence of freeze-on, supporting your observation.

---

## Author Comment (AC2) · 21 Mar 2018

We thank the reviewer for the thoughtful comments and for useful references that have improved the quality of the manuscript. We have addressed the specific comments below (reviewer's comments in bold). We will upload the new version of the manuscript and supplement, with and without changes tracked, separately.

**This paper describes observations of mass budget for the Crosson and Dotson ice shelves in the Amundsen Sea sector of the Antarctic ice Sheet and attempt to provide a narrative for the timing of imbalance and for the succession of events and interaction that led to the current state of imbalance and mass loss of both**

[Figure]

**floating and grounded ice in the region. The paper integrates new and existing observations with modelling in a balanced way in the context of existing literature. The manuscript is well structured, detailed and illustrated. I support the publishing of the manuscript with only minor changes.**

We thank the reviewer for the support of publication and have incorporated the specific comments as described below.

**General comments: I find the title vague and confusing; how varied, what response, and how varied are the elevated melt in the first place. Do you imply here that the same, elevated, melt under both ice shelf induce the various response? From the findings of this paper it does not look to me that this is the case.**

We have changed the title to clarify that the response is in the flow of the ice shelves, and removed "varied," which was unclear: Changes in flow of Crosson and Dotson Ice Shelves, West Antarctica in response to elevated melt

**Thinning rates used to calculated ice shelf elevation back in time are resolved over spatial resolution of 10's of kilometres; recent work have shown that, at least for Dotson, thinning is highly localised to the grounding lines and on the western margin of the shelf. You should at least address the impact that this might have, or not, on your analysis. You allude to it in section 4.3 but can you be more specific as to where this might be the case?**

We think that this most appropriately belongs in the portion of the manuscript that compares our results to previous estimates of the pattern of melt (Gourmelen et al. 2017 and Khazendar et al. 2016). In short, we expect this error affect areas of high melt. We have added the following to section 5.3: "The peak regionally-averaged melt rates found in this study (23 m/yr) occur in the same regions (K1, SW1 and SE1) as found in other studies but the values are lower than the locally computed rates (50 m/yr in Gourmelen et. al (2017) and 129 m/yr in Khazendar et al. (2016)). This difference

is not surprising, since the polygon-based method we use has limited spatial resolution and thus misses variations in melt rate in small-scale features such as within the channel on Dotson. Over broad scales, flux-gate and altimetry-based methods should agree, and indeed the overall melt rate we find beneath Dotson, 7.7±1.3 m/yr, agrees with the 6.1±0.7 m/yr found by Gourmelen et al. (2017). The lower peak melt rates found here compared to prior studies may also result from our assuming a spatially constant thinning rate over shelves where thinning rates vary substantially (Gourmelen et al., 2017). The shelf-wide rates used here likely cause us to underestimate melt in small polygons where thinning is most rapid (e.g. K1, SW1, and SE1 in Figure 3). Similarly, they may cause us to slightly overestimate melt over broad, slower-changing regions (e.g., D1 and D2), but this effect should be smaller due to the shelf-wide thinning rate being more representative of rates in these regions. On scales smaller than the polygons we use to calculate melt, thinning may reach 50 m/yr (Gourmelen et al., 2017), comparable to flux divergence in these areas, causing the polygon-averaged rates to locally underestimate melt by a factor of two and introducing an error in the polygon average that we estimate may be as high as 50% in small polygons near the grounding line. "

**Specific comments**

**Page1**

**L12: Ambiguous, suggests rephrasing. Is what is meant here that the melt under Dotson is dominating the mass loss? Or that the melt under Dotson is driving the mass loss elsewhere? From fig.3 it looks as though the mass loss of Dotson and Crosson are pretty much equal, that the increase in melt rate through the study period is similar, and that one is dominated by loss through basal melt (Dotson) and the other through calving (Crosson).**

The increase in melt is greater on Dotson, but indeed the overall mass loss between the shelves is similar and the change is similar. We have changed the sentence to

"These ice shelves have lost mass continuously since the 1990s, and we find that this loss results from increasing melt beneath both shelves and the increasing speed of Crosson."

**L17: The preceding line gives an explanation for the observation, but this one does not - what is causing the lack of response of Dotson's velocity?** Added: "likely a result of the sustained competency of the shelf."

**Page 2**

**L36: The terminology is indeed varied - Smith West is sharing a drainage basin and an ice shelf with Kohler, shouldn't this then be Kohler East instead as in Mouginot et al., 2014?**

While Smith West and Kohler share an ice shelf, Smith East and West are flowing together as they reach the grounding line. Though there are justifications for either choice, we chose this terminology because the two trunks of Smith merge while grounded, and thus naming as in Mouginot et al., 2014 necessitates drawing a division between Smith and Kohler East in the midst of fast flowing, grounded ice. Following the naming conventions here, divisions between grounded glaciers match natural boundaries in velocity.

**Page 3**

**L15: Ambiguous, suggest: "No similar changes in extent were observed over Dotson"**

Agreed, thanks.

**L23: "over" -> "under"**

Done.

**L24: "much higher than the surrounding ice" with the exception of the grounding line regions of Smith and Kohler** Added the parenthetical "(though lower than at the

grounding lines of Smith and Kohler)"

**Page 5**

**L15: Medley has no sup. Material.**

This is correct, our phrasing was poor. Changed to "details can be found in Medley et al. (2014) and the supplementary materials to Joughin et al. (2014)."

**L27: What air thickness is found here and how does this compares with modelled values? There are evidence that firn-air content found using this approach differs from model output, so well worth addressing this here. A map in Supp. Mat. For example.**

We have added a sentence in the main text to give the range of values we find using this approach and to compare this to the Ligtenberg et al. 2011/RACMO2.3 model output. We have also added a figure showing the spatial pattern of firn-air content in the supplement (Figure S2). The relevant portion of the main text reads: "We find the firn-air content to range between 12-18 m across most of the ice shelves (Supplementary Figure S2); a firn model forced with output from RACMO2.3 shows greater firn-air content than we observe, generally between 20-25 m over these shelves (Ligtenberg et al., 2011)."

**Page 10**

**Line 6: The longer dataset also agrees with recent, high-resolution, estimates.**

We added "Additionally, this longer dataset matches recent, high-resolution radar-based estimates (Gourmelen et al., 2017)."

**Line 10: Reference Shepherd 2010 is missing**

We are unsure of what the reviewer is referring to here: the inline citation is present, and it is listed in the references as "Shepherd, A., Wingham, D., Wallis, D., Giles, K., Laxon, S. and Sundal, A. V.: Recent loss of floating ice and the consequent sea level
contribution, Geophys. Res. Lett., 37(13), n/a-n/a, doi:10.1029/2010GL042496, 2010."

**Page 11**

**L2: You might want to compare and contrast the weakening described here with the shelf weakening and impact on speedup of Smith described in goldberg et al (2016, fig 3) - I think it is relevant here as there are similitudes but also new patterns, especially along the Eastern margins.**

We were unaware that these ice shelves were studied in that publication, and we thank the reviewer for the reference. We chose to address this comment in the discussion of Crosson's speedup, and added: "Prior work has addressed the effect that thinning (equivalently, weakening) over different portions of Crosson and Dotson would have on ice loss upstream (Goldberg et al., 2016), and the areas in which we find weakening encompass several regions that are important for upstream dynamics. That work suggests that while the overall dynamics are insensitive to the bulk of weakening we find here, weakening in key areas, particularly near the Haynes tongue and at the western shear margin, is important for loss of grounded ice upstream, and thus may have influenced speeds near the grounding line as well."

**Page 13**

**L14: There is also no indication of freeze-on from EO-based basal melt extraction**

We thank the reviewer for the comment clarifying this statement. We have added the following: "Previous work (Depoorter et al., 2013; Gourmelen et al., 2017; Rignot et al., 2013) found no evidence of large-scale freeze-on beneath Dotson, and we eliminated the possibility of localized freeze-on being the primary cause of the channel by comparing radar observations to floatation levels."

**L35: There seems to be melting occurring at the southern end of this portion of the channel. In fact it looks to me that the effect of the advection is visible**

**along most of the length of the channel (ice flow is transverse to the channel up
to about half-way down the shelf) as the locus of melt occupies only a fraction of
the entire topographic depression. Then, can other sections of the channel be
used to derive timing?**

We tried to use as much of the channel as possible for this analysis. There are two
factors limiting where this analysis is viable: whether flow was transverse to the channel
and whether there was an obvious point upstream at which to assume the channel
originated. We think our analysis extends as far westward (downward in Figure 6a)
as it can given these two conditions. Because of the along-flow channel coming from
near the grounding line of Kohler immediately westward of where our analysis cuts off
(Figure 6a), we lack an upstream edge of the channel in this region. Moreover, the
westernmost (bottom) line/string-of-pearls in Figure 6a is no longer fully perpendicular
to the channel, and the channel is more aligned with flow farther westward. West of
the outflow channel from Kohler, the channel we analyze is nearly parallel to flow, and
so cannot be used for timing.

**How would ice divergence influence the time-evolution of this channel's width
and depth (e.g. Drews et al., 2016)?**

The difference in orientation of the portion of the channel that we analyze compared to
Drews et al. makes comparison to that work difficult. Drews finds almost no expres-
sion of the channel in the along-flow velocity or strain, though he does find a gradient
across flow (his figure 6). We do not expect an across-flow gradient here because the
channel itself is oriented across flow, and thus the thickness gradient driving the effect
on across-flow strain is absent. We also cannot find such a gradient in the across-
flow surface strain (from the InSAR velocities). In the along-flow direction, there is no
expression in the along-flow surface strain as calculated from the InSAR velocities.
Moreover, we expect the way in which we calculate the timing to be relatively insensi-
tive to ice divergence. Since we integrate the travel time along flow to get the timing, we
are accounting for the along-flow change in ice speed/the along-flow strain. However,

we acknowledge that there may be some effect of along-flow strain not detectible in the velocity, though such an effect must be small. We have added: "While ice divergence following the initiation of channel incision may have had an effect on the channel width (Drews, 2015), there is no change in the divergence of the measured surface velocities around the transverse portion of the channel (conversely, along the flow-parallel portion there is significant convergence). Thus, any change in width due to divergence is at or below the level of uncertainty in the velocity measurements, and therefore small compared to the advection of the channel."

**How does this timing relate to timings deduced from grounded ice thinning and ice shelf channel formation (e.g. Konrad et al., 2017; Gourmelen et al., 2017)?**

While Konrad et al. get tight bounds on the timing of imbalance on Thwaites and PIG, for these glaciers the uncertainties are extremely high, and so unsurprisingly there is significant overlap with the timing we infer. We have added a sentence simply saying that our results are consistent with the timing found by those authors and by Gourmelen et al.: "This timing is consistent with prior work on the upstream propagation of thinning over these glaciers, which found that thinning initiated around 1970ïĆś30 at the grounding lines of Smith and Kohler (Konrad et al., 2017), as well as with an altimetry-based study of Dotson, which found that the shelf had been thinning for at least two decades but not more than a century (Gourmelen et al., 2017)."

**Page 16**

**Line 21: Ok, how does these melt estimates fit in your comparison earlier in this paragraph between melt obtain here and melt rates obtained by Khazendar et al?**

This section was changed substantially in response to comments by the other reviewer. We have made this paragraph a three-way comparison of these studies: "The peak regionally-averaged melt rates found in this study (23 m/yr) occur in the same regions (K1, SW1 and SE1) as found in other studies but the values are lower than the locally

computed rates (50 m/yr in Gourmelen et. al (2017) and 129 m/yr in Khazendar et al. (2016)). This difference is not surprising, since the polygon-based method we use has limited spatial resolution and thus misses variations in melt rate in small-scale features such as within the channel on Dotson. Over broad scales, flux-gate and altimetry-based methods should agree, and indeed the overall melt rate we find beneath Dotson, 7.7±1.3 m/yr, agrees with the 6.1±0.7 m/yr found by Gourmelen et al. (2017). The lower peak melt rates found here compared to prior studies may also result from our assuming a spatially constant thinning rate over shelves where thinning rates vary substantially (Gourmelen et al., 2017). The shelf-wide rates used here likely cause us to underestimate melt in small polygons where thinning is most rapid (e.g. K1, SW1, and SE1 in Figure 3). Similarly, they may cause us to slightly overestimate melt over broad, slower-changing regions (e.g., D1 and D2), but this effect should be smaller due to the shelf-wide thinning rate being more representative of rates in these regions. On scales smaller than the polygons we use to calculate melt, thinning may reach 50 m/yr (Gourmelen et al., 2017), comparable to flux divergence in these areas. This thinning may cause the polygon-averaged rates to locally underestimate melt by a factor of two and introduce an error in the polygon average that we estimate may be as high as 50% in small polygons near the grounding line."

**Fig. 3: b and c should have the same y-axis range (e.g. [0 50])**

We have matched the axes.

**caption: steady state (SS) The cross-hatched light grey bar is labelled floating ice in the figure and grounded ice in the caption - should it be floating ice everywhere?**

Indeed, the cross-hatched bars should be labeled floating everywhere, thanks. We have added parentheticals to define the acronyms "SS" and "GL".

---

## Author Comment (AC3) · 21 Mar 2018

We thank the reviewer for a thorough review that has helped to significantly improve the clarity of the manuscript. Comments from the reviewer (bold) and our responses follow. We will upload the new version of the manuscript and supplement, with and without changes tracked, separately.

**This manuscript presents a detailed analysis of the changes in speed and grounding position of two rapidly changing ice shelves, and their respective catchment basins, in the Amundsen Sea sector: Crosson and Dotson. The work combines a diverse set of measurements: satellite remote sensing data**

[Figure]

**of horizontal velocities and elevation changes, airborne radar sounding for ice thickness, satellite imagery to determine iceshelf front position, and published estimates of ice-shelf thinning rates. The authors then estimate the ice viscosity using a numerical model and perform an analysis of changes in ice-shelf flux, basal melt, and calving rates. Their results show that Dotson and Crosson ice shelves exhibit different dynamic responses to similar perturbations, and that is likely that both ice shelves were out of balance well before the beginning of the satellite record (the 1990s).**

**General comments: The manuscript presents a comprehensive study on the mass balance and velocity state of Dotson and Crosson ice shelves. The work provides some important insights into the origin of the current observed mass imbalance on these ice shelves and their respective catchment basins. I couldn't find major issues with the work. Overall, the manuscript is well written, and the science is solid. I have, however, a few comments/suggestions that I think need further clarification and will improve the presentation of the work. These edits should be straightforward to incorporate.**

**In particular, the Summary section needs a bit more work. I think there should be a preference for the active voice in the 'Summary' (e.g. we analyzed, we found, we concluded, we showed), briefly stating the implications of your findings, as well as summarizing the path that leads to your conclusions. I feel this is missing here.**

We have changed the wording of a number of sentences in the summary to be active and to clarify what work we did and what had been done in prior work. Some examples are: "We find that thinning and speedup. . .", "These conditions lead us to speculate that. . .", "Our results indicate that. . .", and "We used a diagnostic ice-flow model to show that. . ."

**On the melt rate calculation; Fig. S1 shows melt rates estimates for 1-5 year time**

**intervals. Since these are anomalies in basal melting, it is expected (in fact it has been demonstrated, e.g., Jacobs et al., 2013; Dutrieux et al., 2014; Christianson et al., 2016; Paolo et al., 2018) that melt rates are (highly?) sensitive to changes in ocean forcing in the Amundsen Sea sector, which fluctuates substantially at interannual timescales. So, I am unsure how much "weight" one can put on these (highly variable) short-term estimates of basal melt rates in the context of past (longer-term) dynamics of the ice shelf.**

**I am a bit confused about the fact that you used an average thinning rate for the entire ice shelf, which it might be fine for your polygon estimates, but then compare point estimates of basal melt against Khazendar et al. (2016). How meaningful is this comparison? If I understand correctly, the spatial variation in your melt-rate estimate is determined mostly by the flux divergence since you do not account for the spatial variability in thinning rates, is that right? If so, how significant do you expect the changes in dH/dt across the ice shelf to be compared with the spatial changes in horizontal velocity?**

The spatial variation in our melt-rate estimates is almost entirely determined by the flux divergence; the only spatial variability in thinning we account for is slightly different rates for Crosson and Dotson. dH/dt (if the change is assumed to be all ice and not snow or firn) and horizontal flux divergence both map linearly into our estimate of the melt rate. Locally, horizontal flux divergence can vary between ïĆś100 m/yr, while local thinning rates found in Gourmelen et al. (2017) reach as high as 50 m/yr. Our spatially averaged estimates do not capture these peaks, but the estimates of Khazendar et al. explicitly measure thinning and thus, when corrected for flux divergence, ought to represent the anomalous melt at that point. In short, these methods are sensitive to different sources of error, and can provide information on different spatial and temporal scales, so we think that they are complementary and the comparison is worthwhile.

In response to these comments, and suggestions from the other reviewer, we have entirely re-worked the text surrounding this comparison (section 5.3). The section now

includes three paragraphs. The first discusses the limitations of Khazendar et al. study, and how our result complement theirs. The second paragraph discusses the uncertainties and limitations of the Gourmelen study, and why our results are still useful in light of this more spatially resolved estimate. The third reconciles our lower value peak melt with the prior estimates by discussing the effects of spatial averaging and spatial variation in thinning.

**The title needs to be a bit more informative (Observed elevated melt? Over what period? Elevated relative to what? What kind of response?).**

We have changed the title to emphasize the type of response discussed in the paper. It is now: Changes in flow of Crosson and Dotson Ice Shelves, West Antarctica in response to elevated melt

**To facilitate the reviewing process, please use continuous line numbering throughout the entire manuscript, and number every single line.**

We apologize that this numbering makes reviewing more difficult, but we have adhered to the guidelines provided by The Cryosphere.

**Specific comments: Page 1**

**Lines btw 5-10: "remotely sensed datasets". Which datasets?**

Changed to "remotely sensed measurements of velocity and ice geometry"

**Lines btw 10-15: "melt" => "elevated melt" or "melt in excess" (there is basal melt in the steady state)**

Changed to "elevated melt"

**Lines btw 20-25: "instability" => "internal instability"?**

Changed to "susceptible to internal instability triggered by increased ocean melting of buttressing ice shelves." The instability generally requires some triggering (despite the

potential for internal instability, these ice streams may have maintained their position for 1000s of years with no external forcing).

**Lines btw 20-25: "are the dominant source of sea level rise" => "are currently the dominant source of sea level rise" or "are the dominant source of current sea level rise"**

Changed to "currently the dominant source of sea level rise"

**Page 2**

**Lines btw 20-25: "ice flux on ice thickness" => "ice flux across the grounding line on ice thickness there"**

Done

**Lines btw 25-30: "deeper ocean waters are generally warmer in these systems". Careful here, this cannot be a general statement. In locations where CDW is present (not everywhere), there is more of this warm water reaching the deep grounding lines because the CDW is denser than surface waters. This is different from saying "deeper waters are warmer".**

We have revised these sentences to avoid over-generalizing. They now read: "In the case of a retrograde bed, the deepening of the grounding line caused by ungrounding also increases melt because the melting point decreases with depth. For glaciers along the Amundsen Sea, this effect can be intensified because warm, dense circumpolar deep water generally intrudes at depth and results in elevated melt at deeper grounding lines (Jenkins et al., 2016; Thoma et al., 2008)."

**Lines btw 30-35: "recent speed and thickness change" => "recently observed speed and thickness changes"**

Done

**Page 3**

**Lines btw 10-15: "significant changes in the extent of Crosson". How significant? Can you provide a percentage of area change?**

We rephrased this to be more specific. The percentage change in area is deceptive because of extension of the middle of the calving front the absence of any large calving event during this period, so we calculated the absolute area change of the margins. The sentence now reads: "There were also significant changes in the extent of ice at the margins of Crosson ( 250 km2 of ice extent lost) and thus in the amount of contact with its sidewalls and with the tongue of Haynes Glacier (Figure 1b) through this period."

**Lines btw 30-35: "not straightforward" => "challenging"**

Done

**Page 4 Lines btw 30-: How is the uncertainty propagated for the values predicted by the quadratic function where there is no data at all? This uncertainty should increase (substantially) as the predictions get further away from the period constrained by data.**

Added "While we are unable to formally calculate the uncertainty of the surface elevations produced by this extrapolation, we estimate it as 50% of the change from the earliest measurement (in 2003)."

**Lines btw 30-: Not sure I understand the following statement: "The results from this method more closely match the available ICESat-1 data than using the elevation and thinning rate from 2004 alone, as was done in Mouginot et al. (2014)". In what sense it matches better ICESat (2003-2009), which is outside of the extrapolated range (1996- 2002)?**

Expanded this sentence to: "To assess relative to previous methods, we calculated estimated surface elevations for 2003-2008 using this quadratic function to test its ability to match the available ICESat-1 data. Residuals are smaller than those resulting from

using a quadratic fit to the elevation and thinning rate from 2004 alone, as was done in Mouginot et al. (2014)."

**Page 5**

**Lines btw 5-10: "The thinning rates on the grounded ice are more accurate since they lack tidal effects". This is not the main reason why altimetry thinning-rate estimates may be more accurate over grounded ice. Over (rapidly-changing) grounded ice the signal-to-noise ratio is much higher than over floating ice where the altimeter only "see" about 10% of the thickness-change signal due to hydrostatic balance. Also, tide models are reasonably good (outside of the grounding zone) at the current stage.**

Indeed, this effect is large. We have update the text to: "The thinning rates on the grounded ice are more accurate because the entirety of any change to thickness is manifest in the surface elevation while over floating ice 90% of the thickness change is accommodated through raising the ice bottom due to hydrostatic balance. Thus, we smoothed the thinning over the shelves for 10 km downstream of the grounding line to preserve continuity and reasonable surface slopes."

**Lines btw 15-20: "This method reduces many of the artifacts that can occur when interpolating sparse ice thickness measurements while avoiding making any assumptions about the present state of balance (e.g. assumptions for mass conservation methods (Morlighem et al., 2011))". It is still interpolation, right? Meaning that regardless of any "smart" weighting, you don't have information in between the sparse thickness samples. The mass conservation approach uses additional information (velocities) to fill in those gaps. Bottom line, one interpolation method, another interpolation method... it is still interpolation.**

We were primarily concerned with pre-empting the criticism that we should have been using a mass-conserving bed. We have updated the text to clarify that this is just a different interpolation method. We added: "though like other methods used to interpolate

radar data it still has high uncertainty due to the sparseness of the underlying radar profiles."

**Page 7**

**Line 1: "$m_s$ is the basal melt rate". $m_s$ => $m_b$**

Thanks, fixed.

**Line 5: I think a more appropriate reference for RACMO2.3 SMB is "Van Wessem et al. (2014)".**

We have updated the text and figure to use the 20% error that Van Wessem et al. estimate rather than that used in Depoorter. This sentence now reads "For the SMB, we use the annual mean for 1979-2013, which has an uncertainty of $\pm$20% (Van Wessem et al., 2014)."

**Page 8**

**Line 16-17: "but this choice should not have affected the overall spatial pattern". How sure can you be about this (i.e. what is the effect of many small artifacts on the overall result)?**

We cannot be positive, but this would generally be more worrisome if we had found areas with strengthening and weakening intermingled, where regularization may have forced an entirely different solution. Rather, we find areas in which we infer weakening, and expect that regularization would just diffuse these areas. We have updated the text to read "The lack of regularization may have concentrated the weakening or strengthening into smaller areas than would have been found with regularization, but any solution, regularized or not, likely would have to introduce weakening into these same areas in order to reproduce velocity field. Thus, the lack of regularization likely did not affect the overall spatial pattern of weakening."

**Page 9**

**Line 9: "and any additional amount entering the shelves in each year". Suggest adding "by precipitation over the ice shelves" or "through ice-shelf surface accumulation".**

We have clarified the three categories of incoming flux that we are partitioning. This sentence is now "The incoming flux consists of ice-shelf surface accumulation and the flux across the grounding line; we partition the grounding-line flux into a steady-state amount (i.e. the accumulation in the catchment upstream) and any additional amount (in excess of steady state) entering the shelves in each year."

**Page 10**

**Lines 12-13: "would not substantially alter our qualitative conclusions". You can remove "qualitative" here.**

Done

**Lines 8-9: "implying that we likely underestimate the error in our melt and flux calculations". Not necessarily. The discrepancy in the thinning rates on those studies is mostly due to some using a 5-year (laser) altimetry record while the other using an 18-year (radar) altimetry record. So, the different thinning rates are likely representing different timescales. For example, as shown more recently by Paolo et al. (2018), there are large fluctuations in ice-shelf thickness at ENSO timescales (âĹij4-5 years), a time span comparable to the ICESat period.**

Indeed, there may be real variability incorporated into these differences, but in the case of Dotson it is almost certainly error. Paolo et al. 2015 show the trend in thickness in the extended data, and it is very nearly linear in the longer timeseries. Since this longer record encompasses the time covered by the laser record, the factor of 2 difference in estimated thinning rate is not just a sampling effect on real variations, though perhaps some of it is. We have updated the text to reflect the possibility that a component of the discrepancies results from variability. The relevant text now reads: "The thinning

rates measured via radar altimetry use a longer time series of thickness data (Paolo et al., 2015), and so we expect these values to be more representative of the average thinning over our study period than previous laser-altimeter based estimates, which range from 36-63 Gt a-1 (Depoorter et al., 2013; Pritchard et al., 2012; Rignot et al., 2013; Shepherd et al., 2010). Some of the range in measured thinning may reflect real multi-annual variability (Paolo et al., 2018) that is sampled differently during the 5-year laser-altimetry record compared to the 18-year radar-altimetry record. However, even if this discrepancy reflects real variability, we likely underestimate the error in our melt and flux calculations by using a temporally constant thinning rate and propagating only the stated error from Paolo et al. (2015). While using different thinning rates substantially alters the estimated melt, melt rates on Dotson calculated using any of these values are larger than the grounding-line flux. Thus, using a different thinning rate within the range of published values would not substantially alter our conclusions, though it would imply greater magnitude of melt."

**Page 11**

**Line 5: "ice that is thinner and cooler than expected" => ???**

Added "This effect is likely not real, but rather is introduced by the model as compensation for poor estimates of temperature and thickness at the calving front."

**Page 13**

**Lines 13-14: "the downstream increase in thickness is large compared to the SMB". Can you provide the typical magnitude of SMB over this region (e.g. from RACMO) to put things in context?**

Done. Rates from RACMO range from 0.48 to 1.21 m/yr.

**Page 16**

**Lines 16-24: When estimating basal melt rates using ice-shelf surface height changes (e.g. estimates derived from altimetry), there is a substantial uncer-**

**tainty associated with fluctuations in surface mass balance, which can have
large amplitudes at seasonal-to-interannual timescales. Moreover, it has been
shown that, often, increase basal melting and surface height change are out of
phase (i.e. they can go in opposite directions) (e.g. Paolo et al., 2018).**

We have reworked this section in response to the general comments of this reviewer,
though this point remains relevant. We have added the following: "While Gourmelen et.
al (2017) are able to compute spatially resolved melt rates beneath all of Dotson, their
altimetry-based method has greater sensitivity to certain errors than the methods we
employ. The amount of snow on an ice shelf significantly influences surface elevations
because the lower-density snow does not hydrostatically depress the shelf as much
as an equivalent thickness of ice. Thus, uncertainty in SMB leads to significant uncer-
tainty in thickness changes, particularly because SMB may be inversely correlated with
basal melt on seasonal-to-interannual timescales (Paolo et al., 2018). Moreover, mis-
measurement of the surface elevation is increased tenfold in estimating the melt rate
using altimetry-based methods, leading to substantial uncertainty. Our method is pri-
marily sensitive to horizontal flux divergence, so it is less sensitive to errors in surface
elevation and SMB than the method of Gourmelen et al."

**Line 28: "similar" => comparable**

Done

**Line 28: "changes in melt" => changes in basal melt**

Done

**Line 30: "were associated" => by whom?**

We changed this sentence to the active voice: "We find that thinning and speedup early
in the study period are likely an ongoing response to earlier changes."

**Line 32: "total basal melt was further increased" => by whom? Please avoid
the passive voice. It is not clear whether you are stating known facts (from the**

**literature) or stating your conclusions.**

We clarified what has been established previously and made this sentence active: "Similar to previous studies, we show that basal melt rates increased on areas that were floating throughout the study period, and we find that total basal melt was further increased as ungrounding exposed more area to melt. "

Line 34: "A change in ocean forcing years or decades before 1974 likely led to in Dotson's imbalance in 1996". This is highly speculative, particularly regarding "a change in ocean forcing". It is OK to speculate but clarify (in the Summary) what led you to this conclusion. We amended this sentence to make explicit that this is our speculation. The sentence is now: "These conditions lead us to speculate that change in melt, likely resulting from a change in ocean forcing years or decades before 1974, may have led to in Dotson's imbalance in 1996. "

**Page 17:**

**Lines 4-6: Are you suggesting this as future work? If so, please clarify you are pointing the future direction of work needed; and justify why we would need such direction (i.e. what's the relevance in the context of understanding and predicting future ice-shelf/icesheet loss).**

We have added an additional sentence to state the importance and clarified that we are suggesting this as future work. The last lines now read: "Determining the initial cause of change to this system is key to understanding whether the present retreat results from ongoing oceanic or climatic changes, natural variability, or internal instability, and thus important for placing these observations in the context of other changes to submarine basins around Antarctica. In the future, prognostic modeling of this system beginning in 1996 or before (i.e. "hindcasting"), could help test how different initial perturbations to the system would have affected its flow speed and mass balance, and thus provide context to these changes relative to those observed in other glaciers."

**Supp. Page 2: Line 18: "and we are unsure of the exact method used by Khazendar et al". Why don't you ask them?**

We emailed Ala Khazendar, and he indicated that he thought this point anomalous. Those authors had referred to an anomalous point in the text, though the connection between that portion of the text and this particular measurement had not been clear to us. We thus eliminated the speculation that interpolation caused this discrepancy and note only that they identify that point as anomalous. The relevant text now reads "...our values for the thinning rate agree to within error for the 27 points that they do not identify as anomalous (Table S2)." Additionally, we have added a note to the caption of table S2 stating: "Note that Khazendar et al. identify point 24 as anomalous, though the value we calculate is similar to others in the area."